# Stony coral tissue loss disease decimated Caribbean coral populations and reshaped reef functionality

Lorenzo Alvarez-Filip [1,✉], F. Javier González-Barrios[1], Esmeralda Pérez-Cervantes[1], Ana Molina-Hernández[1] & Nuria Estrada-Saldívar[1]

Diseases are major drivers of the deterioration of coral reefs and are linked to major declines in coral abundance, reef functionality, and reef-related ecosystems services. An outbreak of a new disease is currently rampaging through the populations of the remaining reef-building corals across the Caribbean region. The outbreak was first reported in Florida in 2014 and reached the northern Mesoamerican Reef by summer 2018, where it spread across the ~450-km reef system in only a few months. Rapid spread was generalized across all sites and mortality rates ranged from 94% to <10% among the 21 afflicted coral species. Most species of the family Meandrinadae (maze corals) and subfamily Faviinae (brain corals) sustained losses >50%. This single event further modified the coral communities across the region by increasing the relative dominance of weedy corals and reducing reef functionality, both in terms of functional diversity and calcium carbonate production. This emergent disease is likely to become the most lethal disturbance ever recorded in the Caribbean, and it will likely result in the onset of a new functional regime where key reef-building and complex branching acroporids, an apparently unaffected genus that underwent severe population declines decades ago and retained low population levels, will once again become conspicuous structural features in reef systems with yet even lower levels of physical functionality.

[1] Biodiversity and Reef Conservation Laboratory, Unidad Académica de Sistemas Arrecifales, Instituto de Ciencias del Mar y Limnología, Universidad Nacional Autónoma de México, Puerto Morelos, Quintana Roo, México. ✉email: lorenzo@cmarl.unam.mx

Disease outbreaks are often associated with mass mortality events that can rapidly and drastically reduce populations over short periods of time[1–3]. When these events affect foundation species, population losses result in changes in the local environment, upon which a variety of other species depend, and altering the structure and functioning of the entire ecosystem[4,5]. In the marine realm, Caribbean coral reefs offer some of the best examples of such catastrophic events. This region is a well-known hot spot for diseases[6–8] and many have decimated the populations of primary reef-builders, resulting in devastating changes to the spatial heterogeneity of the reefscape, its capacity to provide ecosystem services, and the ability to track sea-level rise[6,9]. In the late 1970s, a region-wide outbreak of white-band disease led to population losses of nearly 80% of the major reef-building corals *Acropora palmata* and *A. cervicornis*[10], resulting in a notable reduction in reef functionality (e.g.,[11–13]). Since then, multiple disease outbreaks have nearly decimated the populations of many other key reef-building corals[6,7]. Moreover, there is an ever-increasing risk of diseases that may further impact the stability of reef ecosystems, as the frequency and intensity of disease outbreaks are often related to rapidly increasing human-induced pressures, such as rising sea temperatures, decreased water quality, and nutrient enrichment[14–18].

In 2014, the south-eastern Florida sub-region saw the onset of a new deadly coral disease, known as stony coral tissue loss disease (SCTLD;[19]). The disease spread across the Caribbean with reports of SCTLD in the Western Caribbean, Bahamas, Puerto Rico, and the US Virgin Islands and recent reports in the Lesser and Greater Antilles[20]. The sources of transmission and specific causative agents are not yet fully understood, although the disease is clearly transmitted through seawater with bacteria being involved at some level in disease progression, and viruses of the algal symbionts being found in pathological studies, suggesting a disruption of host–symbiont physiology[21–23]. Most evidence of the devastating effects of SCTLD comes from Florida and Mexico[24,25], yet similar outcomes have been reported elsewhere in the region. SCTLD affects nearly 30 different coral species and is highly virulent, spreading rapidly within reef systems[26,27]. Highly susceptible species (e.g., *Meandrina meandrites* and *Dendrogyra cylindrus*) are rapidly infected and can die within weeks[19,28]. Coral density and composition and environmental conditions, including nutrient concentrations and turbidity, likely influence disease prevalence and progression[24,25,29,30]. However, high water temperatures do not appear to directly influence SCTLD prevalence or virulence[21,24,26,27].

SCTLD now threatens many coral species that serve as important reef-builders and habitat providers in most of today's Caribbean reefs[31], which raises the question of whether post-SCTLD coral assemblages will be able to maintain key geo-ecological functions, such as reef framework production, sediment generation, the maintenance of reef habitat complexity, and the capacity for coral reef growth that is needed to track future sea level increases[9]. These functions largely rely on the capacity of coral assemblages to create complex three-dimensional structures by means of calcium carbonate precipitation – defined hereinafter as reef physical functionality. This is particularly relevant in the Caribbean, as lower levels of functional diversity are present compared to those in other regions (e.g., the Great Barrier Reef). Moreover, many key functional attributes of reef-building corals are redundant, and only a few species determine the physical functionality of the coral community[32,33]. Here, we used extensive pre- and post-SCTLD data along a 450-km reef track in the Mexican Caribbean to examine the regional effects of this emerging threat and to identify the potential ecological and environmental covariates that influenced the occurrence and severity of SCTLD. We then sought to answer how the severe

population declines of SCTLD-afflicted species would impact functional diversity and coral community calcification in Caribbean coral reefs. We characterize the functional diversity of Caribbean coral species using six different traits known to be important for reef physical functionality[32,33] (i.e., skeletal density, growth rate, rugosity index, colony size, reproduction strategy, and corallite width) and show that coral mortality was widespread, with corals with intimate phylogenetic (i.e., Meandrinadae and Faviinae) and functional trait relationships being disproportionally affected by SCTLD, favoring the domination of coral species that poorly contribute to reef functionality.

## Results and discussion

**Impacts on coral communities.** In only a few months, SCTLD spread across hundreds of kilometers and triggered an unprecedented loss of corals (Figs. 1, 2). Out of the 29,095 colonies surveyed between July 2018 and January 2020, 17% were already dead with signs of recent mortality (e.g., bare skeletons or the presence of a thin layer of filamentous algae covering the colony; Fig. 1e) and an additional 10% were afflicted by the disease (Fig. 1c, d). However, susceptibility and mortality varied greatly between species. Twenty-five of the 48 recorded species were affected by the disease, with disproportionate effects observed in a single morpho-functional group largely defined by massive species with mid-to-large sizes, dense skeletons, low growth rates, and broadcasting sexual reproduction (Fig. 1). Species from the family Meandrinadae and subfamily Faviinae were the most severely affected. In particular, *Dendrogyra cylindrus* and *Meandrina spp.* experienced disease prevalence and population losses greater than 80% (Fig. 1). A temporal comparison of community compositions with pre- and post-outbreak data revealed that less conspicuous species, such as *Dichocoenia stokesii* or those within the Mussinae subfamily, were noticeably less abundant after the outbreak (Fig. S1), indicating that these species were more severely affected than what our post-outbreak surveys suggested. This is likely because the skeletons of the small and encrusting species killed by the disease were rapidly covered by algae or sediment, rendering them inaccessible during post-outbreak surveys.

Across all surveyed sites, disease prevalence (considering both diseased and dead colonies) in highly susceptible species showed no statistical differences with regard to depth, reef zone, structural complexity, or coral density (Fig. 3a; Table S1; Fig. S2), suggesting that the spread of SCTLD is primarily controlled by the capacity of the pathogen(s) to be transported in the water column within and between reef sites. However, we found a strong effect of the threat of coastal development on disease prevalence, indicating that sites close to developed areas were considerably more affected than those in isolated regions. In addition, wind exposure (i.e., windward areas) and the age of marine protected areas were also related to higher disease prevalence (Fig. 3a; Table S1; Fig. S2). The observed significant effects were strongly driven by the lack of SCTLD in the reef sites of Banco Chinchorro (which remained unaffected until at least December 2021). Banco Chinchorro is an isolated offshore bank with restricted access and minimal human infrastructure (i.e., fisher campsites and research and military stations) that is separated from the mainland by a deep-water channel in which the strong northward Yucatan Current likely acts as a physical barrier to biological connectivity and land-based perturbations. Species susceptible to SCTLD are abundant in Banco Chinchorro (Fig. 2b), thus the absence of the disease is not due to the lack of potential host species.

When the reefs of Banco Chinchorro were removed from the analyses, high-level coastal development remained significant,

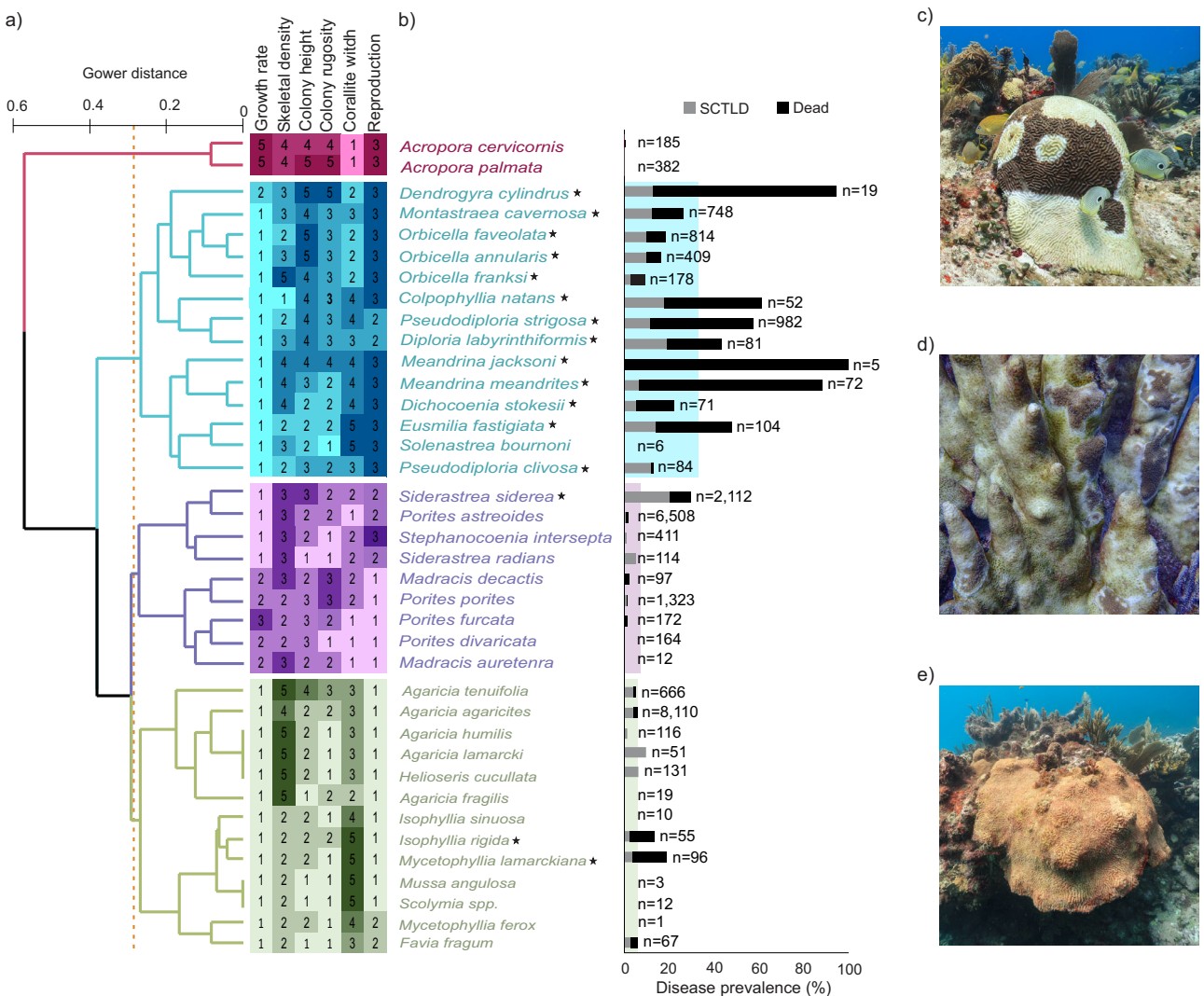

**Fig. 1 Morpho-functional groups of Caribbean corals and their susceptibility to stony coral tissue loss disease (SCTLD). a** Hierarchical clustering dendrogram based on a Gower dissimilarity analysis and heat map representation of functional traits. Numerical values from 1 to 5 correspond to the categories listed in Table S6. The variation in color intensity within each group (light to deep) corresponds to the trait numerical value (given inside each square). **b** Prevalence of SCTLD for coral species across 101 reef sites in the Mexican Caribbean ($n =$ number of colonies). We included coral colonies with total mortality whose deaths could be attributable to SCTLD. The shaded area corresponds to the prevalence of SCTLD for each morpho-functional group in (**a**). The asterisk (*) indicates species with >10% disease prevalence (these were considered highly susceptible species, see Methods for more details). **c** *Pseudodiploria strigosa* colony with the characteristic lesions produced by SCTLD in Puerto Morelos (July 2018). **d** *Dendrogyra cylindrus* colony afflicted with SCTLD showing extensive recent mortality in Sian Ka'an (September 2018). **e** A recently deceased colony of *Meandrina* sp. (covered by a homogenous, thin layer of filamentous algae) in Akumal (September 2018). Photo credits: Lorenzo Alvarez-Filip.

whereas the rest of the covariates ceased to have significant effects (Fig. 3b; Table S2; Fig. S3). The effects of coastal development and wind exposure might be linked and partly explained by the fact that the reefs adjacent to the mainland (all windward sites) were also closer to the main urban and tourist developments in the region. For example, most sites experiencing disease prevalence >50% are located in the northernmost region, where coastal development is high (Fig. 2;[34]). Coastal reefs in the Mexican Caribbean are strongly influenced by freshwater inflow from combined runoff and submarine groundwater discharges, which load the seawater with pathogens, sediments, nutrients, and pollutants[35,36], many of which have been found to serve as vectors and influence disease progression while reducing coral resistance (e.g.,[15,16,37–39]). These negative effects may be exacerbated by low water recirculation typical of reef lagoons in fringing reefs, which increases the influx and retention of coastal waters within the system[35].

Overall, the spatial pattern of afflicted reefs suggests that anthropogenic pressures in the form of coastal development modulate the vulnerability of coral communities to a certain degree, resulting in greater or lesser disease prevalence. Therefore isolation from anthropogenic pressures might provide some degree of protection. However, given the contagiousness and the high mortality rate after infection, coral mortality and the disappearance of key reef-building species will most likely occur regardless of local-scale conditions and geomorphological features after SCTLD reaches a site (Fig. 2;[19,25,31,40]).

**Impacts on functional integrity.** Abrupt coral die-off radically reduced the abundance of species and the traits that support the physical functionality of coral reefs. Most of the reefs shifted further away from the dominance of reef-building species that are key providers of three-dimensionality to depauperate assemblages

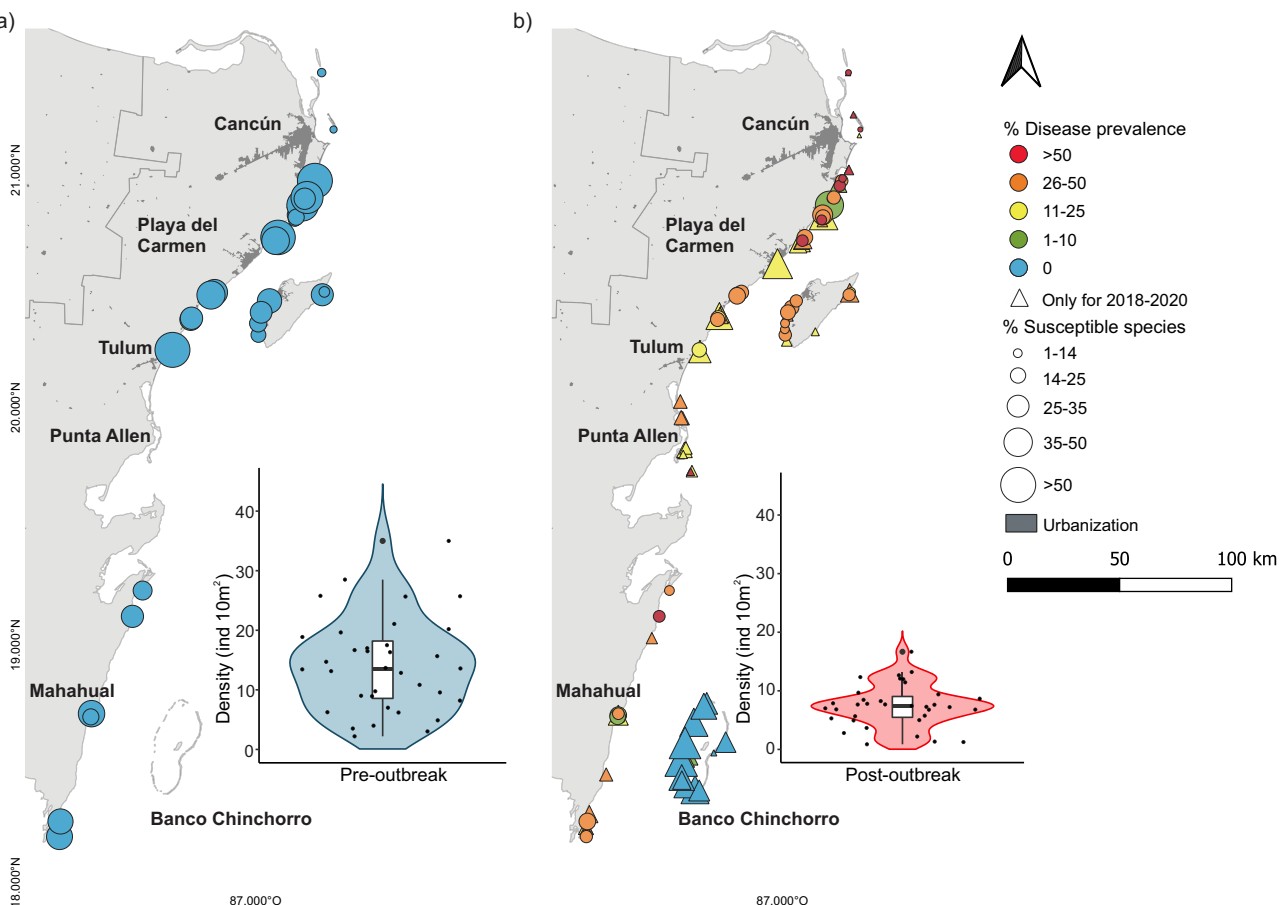

**Fig. 2 Prevalence of stony coral tissue loss disease (SCTLD) in highly susceptible species in the Mexican Caribbean. a** White plague-type disease prevalence in species highly susceptible to SCTLD in 35 sites surveyed before the outbreak of the disease (2016 and 2017). **b** SCTLD prevalence (i.e., diseased and recently deceased colonies) in highly susceptible species during or after the SCTLD outbreak (2018 and 2020). In (**a**) and (**b**), the circles represent the locations of the reefs that were sampled before and after the outbreak. The triangles represent the sites that were only surveyed during or after the outbreak. The size of the figure indicates the percentage of healthy colonies of highly susceptible species based on the total number of surveyed colonies at each site and time period (including all coral species). The insets in (**a**) and (**b**) represent the distributions of the densities of live coral colonies of highly susceptible species across all surveyed sites for each period. The data points represent each surveyed site, and the box plots depict the median (horizontal line), the first and third quartiles (box height), and 95th percentiles (whiskers). The shaded area (violin plot) depicts the kernel density showing the probability of the data at different values. Highly susceptible species are those with more than 10% disease prevalence (see Methods and Fig. 1).

dominated by taxa with simpler morphological attributes and slower growth rates. The results of the similarity percentage (SIMPER) analysis show that even before the outbreak, most coral reefs in the region were already largely dominated by encrusting and sub-massive agaricids and *Porites astreoides*, which are weedy coral species that accounted for 63.33% and 71.83% of the similarity between sites before the outbreak and after the coral die-off, respectively (Fig. S4). The relative increase in the abundance of these two groups accounted for 50.42% of the dissimilarity between periods (pre- and post-outbreak), while decreases of highly susceptible species accounted for only 13.06% of the dissimilarity, as many of these were either uncommon or rare species (Table S3).

Given that the species that suffered the most severe losses share key life-history traits (Fig. 1), the functional space of the coral assemblages was considerably reduced at regional scales after the outbreak. A before-after analysis at species and family levels and the multi-dimensional trait space of the coral assemblages weighted by the absolute abundance of taxa contributing to each trait revealed a drastic transformation towards more homogenous assemblages, as determined by taxonomic and functional trait data, with a notorious lack of contributions from the most severely afflicted species during the post-outbreak period

(Fig. 4a–c). These emerging novel assemblages were remarkably characterized by the presence of acroporid corals and their life-history traits (Fig. 4a–c) despite their low abundance across surveyed sites (Table S3). Not surprisingly, composition changes resulted in significant losses of functional richness (t = 2.67, df = 46.04, p = 0.01) and functional evenness (t = 3.81, df = 65.54, p > 0.01) in the coral assemblages (Fig. 4d, e) despite an apparent increase in species richness (t = −2.36, df = 65.19, p = 0.02) that resulted from the increased survey effort during the post-outbreak period (Fig. S5; see methods). Ultimately, increased mortality was reflected in a marked reduction in coral community calcification (regional mean ± SE; $4.60 \pm 0.77$ G = Kg $CaCO_3$ $m^2$ $yr^{-1}$ before the outbreak to $3.27 \pm 0.53$ G after the outbreak; t = −3.0, df = 34, p = 0.004; Fig. 3f) that was largely driven by the loss of highly susceptible species ($3.04 \pm 0.62$ G pre-outbreak to $1.91 \pm 0.34$ G post-outbreak).

The ecology and physical functionality of coral assemblages in the Caribbean were undergoing severe ecological changes prior to the SCTLD outbreak. Chronic and acute disturbances had progressively driven a decline in the abundance of the main reef-building corals that was accompanied by a concomitant increase in the relative or absolute abundance of opportunistic species characterized by small-sized colonies that do not notably

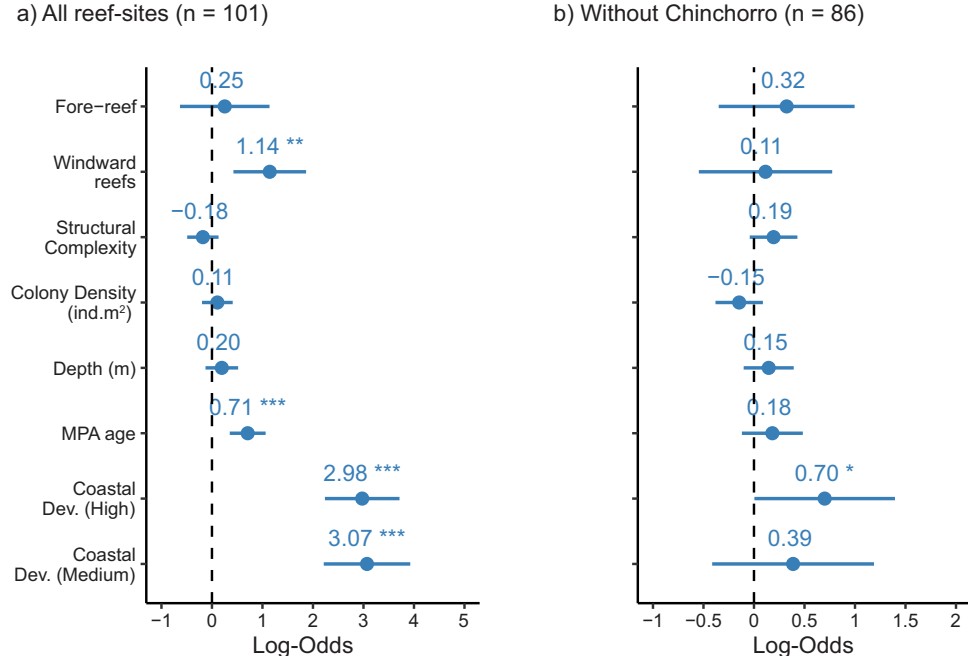

**Fig. 3 Disease prevalence predictors in coral reefs in the Mexican Caribbean. a** Disease prevalence predictors for 101 reef sites in the Mexican Caribbean and (**b**) without the Banco Chinchorro reefs (86 sites). Effect sizes are the logistic mixed models with the dots and lines representing the means and 95% confidence intervals, respectively. All continuous predictive variables were scaled to z-scores with the *scale* function in R. In the model, categorical variables, such as coastal development (low), leeward reefs, and back-reefs, were used as arbitrary references. Asterisks denote statistical differences (*$p <$ 0.05, **$p <$ 0.01, ***$p <$ 0.001).

contribute to reef structure and are known to be tolerant to environmental stress (Fig. 5;[9,32]). The pre-SCTLD communities were described as 'shifted' coral assemblages, and the contributions of formerly dominant acroporids were often negligible given their reduced abundance, whereas large massive species remained and contributed the most to ecosystem structure and functionality (Fig. 5;[32,41–43]). However, the resulting wide-spread coral mortality described here was dictated by the vulnerability to SCTLD of some species that share key morpho-functional traits (Fig. 1a), and thus caused non-random changes in community structure that further and radically affected the functional integrity of the coral communities.

The morpho-functional groups comprised of large and massive species were the most afflicted by the SCTLD outbreak (Fig. 1), whereas the species mildly affected by the disease showed relative increases in abundance. The post-SCTLD coral communities are now represented by a hyper-domination of opportunistic corals, although this remarkably seems to be accompanied by an apparent resurgence of acroporids as key functional elements (Figs. 4a–c, 5). However, the increase of acroporids is primarily an artefact of the drastic reductions in the relative contributions of many other species due to SCTLD (Fig. 5). Only a minor proportion of the increase in the contributions of acroporids may be explained by population recovery or re-sheeting growth over relict reef structures[44–46]. In fact, the acroporid populations have remained low compared with their historical estimations[12,13], as these species have low biological connectivity, low recruitment rates, reduced genetic diversity, impaired recovery abilities, and high vulnerability to regional and global stressors[13,47]. Although it is encouraging that acroporid populations have remained relatively unchanged after the SCTLD outbreak, it is unlikely that these species will considerably improve the structure and dynamics of rapidly changing Caribbean coral assemblages given current conditions.

The large-scale loss of the functionally important corals defined radical shifts in reef conditions and dynamics, exacerbating

further losses of ecological integrity along the entire reef track. The outcomes of coral die-off from the SCTLD outbreak will compromise key functions that are supported by living reef-building corals such as reef framework production, the maintenance of reef habitat complexity, and the potential for growth[9]. These functions largely depend on the capacity of coral assemblages to accumulate calcium carbonate at higher rates than the rate of loss due to biological, chemical, or physical erosion. If erosive processes equal or exceed reef carbonate production, reef frameworks may be destroyed faster than they are produced, resulting in a net negative carbonate budget[48,49]. In this study, we observed a nearly 30% reduction in the capacity of coral communities to produce calcium carbonate. This is alarming because levels of community calcification prior to the impacts of SCTLD were substantially below the optimal rates that have been reported under high coral cover states in the Caribbean[49,50]. In the absence of recovery, the ultimate consequences of coral mortality will thus be modulated by destructive forces like bioerosion or the biogenic dissolution of reef structures[51]. This is particularly relevant as both bioerosion rates and skeletal dissolution are thought to become pervasive when the water chemistry changes or the temperatures increase[48,51]. Our understanding of how the increased availability of substrate for bioeroders will interact with rapid environmental changes remains limited. However, if the ultimate objective is preserving coral reef functioning and services, it may be necessary to focus on replenishing and favoring the recovery of coral communities while improving our understanding of how to control and modulate the destructive forces operating within coral reefs.

**Implications for conservation.** The widespread coral die-off associated with SCTLD has affected the populations of many important reef-building species (Fig. 1). Although the impacts of this highly virulent disease are consistent across affected

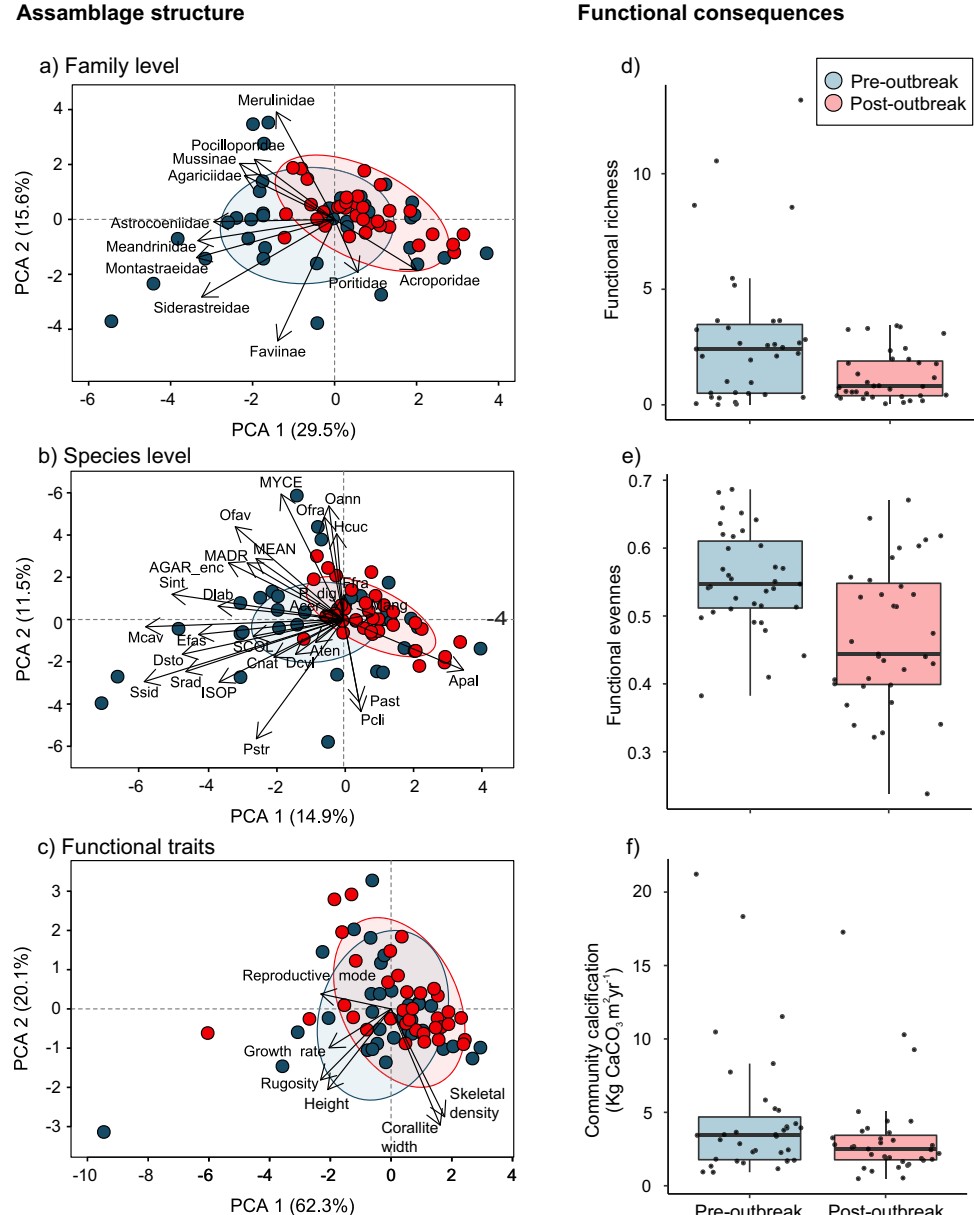

**Fig. 4 Shifts in coral community composition and functioning following the stony coral tissue loss disease (SCTLD) outbreak. a–c** Principal component biplots of the shifts in the coral assemblages in 35 reef sites along the Mexican Caribbean between the pre- (blue) and post-outbreak (red) periods in (**a**) coral assemblages, (**b**) family assemblages, and (**c**) functional traits based on community weighted means (CWM). The points in (**a–c**) represent the reef sites for each period. Vectors represent the absolute contributions of families, species, and traits. The colored ellipses represent the 95% confidence intervals around the weighted average of the site scores for each period. **d–f** Box plots of the functional shifts between the pre- (blue) and post-outbreak (red) periods in the same reef sites with regard to (**d**) functional richness, (**e**) functional evenness, and (**f**) coral community calcification (Kg CaCO3 m² yr⁻¹). The points in (**d–f**) represent each surveyed reef site and box plots show the median (horizontal line), first and third quartiles (box height), and the minimum and maximum values (whiskers; excluding outliers). Species key in (**b**): Acer: *Acropora cervicornis*; AGAR_enc: *Agaricia* encrusting Apal: *Acropora palmata*; Aten*: Agaricia tenuifolia*; Cnat: *Colpophyllia natans*; Dcyl: *Dendrogyra cylindrus*; Dlab: *Diploria labyrinthiformis*; Dsto: Dichocoenia stokesii; Efas: *Eusmilia fastigiata;* Ffra: Favia fragum; Hcuc: Helioseris cucullata; ISOP: *Isophyllia spp*; MADR: Madracis spp; Mang: *Mussa angulosa*; Mcav: *Montastraea cavernosa*; MEAN: *Meandrina* spp; MYCE*: Mycetophyllia* spp; Oann: *Orbicella annularis*; Ofav: *Orbicella faveolata*; Ofra: *Orbicella franksi*; P_dig: Branching *Porites;* Past: *Porites astreoides*; Pcli: *Pseudodiploria clivosa*: Pstr: *Pseudodiploria strigosa*: SCOL: Scolymia spp; Sint*: Stephanocoenia intersepta*; Srad: *Siderastrea radians*; Ssid: *Siderastrea siderea*.

regions[19,24–27], the wide-spread consequences of this outbreak are yet to be known for the entire Caribbean. However, the rapid movement of the disease across the region[20] and the overlapping distribution ranges of most species within the Greater Caribbean region[33,52], suggest that the outbreak will affect the entire region, as has occurred with previous disease outbreaks[6,8,10]. Therefore, some species will rapidly be at a clear risk of extinction across their distribution ranges (e.g., *D. cylindrus;* Fig.1a;[28]), while other

susceptible species that underwent comparatively lower declines (30–70%; e.g., brain corals) will also have compromised abilities to overcome future sources of stress. For example, the evident declines in the populations of species belonging to the family Meandrinidae and subfamily Faviinae could reduce their levels of genetic diversity (e.g.,[53]), putting these species at risk of bottle-neck events that would limit their ability to cope with environ-mental change[54]. In addition, the levels of isolation of the

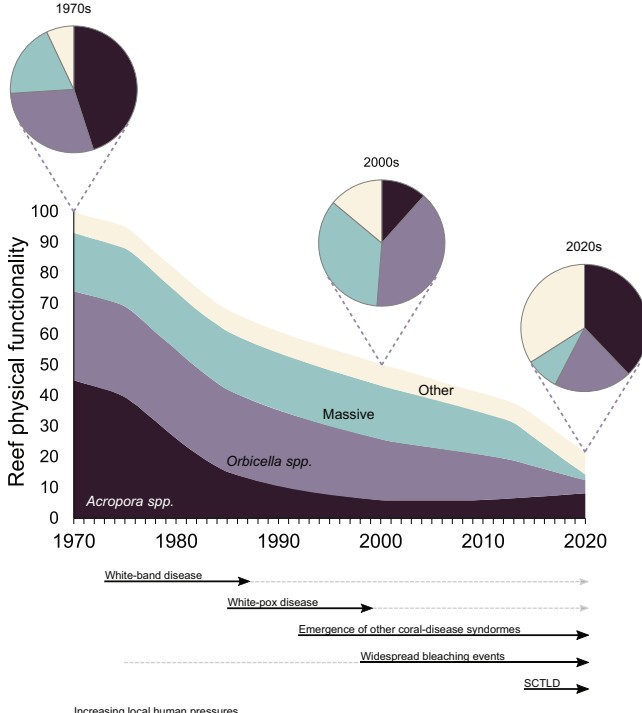

**Fig. 5 Conceptual diagram of the long-term trajectory of the physical functionality of Caribbean reefs based on published temporal trends (Table S4) and the recent impacts of stony coral tissue loss disease (SCTLD).** The physical functionality of reefs depends on the abundance (or cover), capacity to accumulate $CaCO_3$, and structural complexity of each species present in the system[32]. The stacked plot represents the functional contributions of four coral groups over time. The pie charts illustrate the proportional contributions of each coral group during three different periods. *Acropora* spp. and *Orbicella* spp. contain all the species for each of these genera and are illustrated as a single group, as they are the main reef-building corals in the Caribbean and have dominated shallow-water coral-reef habitats throughout the region in geological times[42]. The group of massive corals includes important reef framework builders from the *Diploria*, *Pseudodiploria*, *Colpophyllia*, *Montastraea*, and *Dendrogyra* genera (many of which were severely affected by SCTLD and were included in the second morpho-functional group from the top in Fig. 1a). The other group includes all other coral species, which are largely classified as weedy, submassive, or foliose-digitate corals (included in the third and fourth morpho-functional groups from the top in Fig. 1a) for which little evidence of declines exists. The black arrows indicate major sources of coral decline widely recognized in the literature. White-band disease resulted in severe population declines of acroporids[10]. The white-pox epidemic has infected many of the remaining colonies of this genus since the 1990s[87]. Other coral-disease syndromes (e.g., white plague and Caribbean yellow band) that mainly affect *Orbicella* and other massive species have increased in frequency and virulence over the last three decades (e.g.,[7,88]). Coral mortality has also continued to increase in the Caribbean and is associated with warm-water bleaching events and other local-scale anthropogenic impacts[13,34,89]. The grey-dashed arrows indicate that the source of stress remains, although the effects on widespread coral mortality are unclear.

remaining colonies will reduce or hinder the capacity for sexual reproduction (e.g., Allee effect) and genetic recombination, further diminishing the abilities of populations to adapt to rapidly changing conditions[55]. Moreover, many afflicted species are slow growing (Fig. 1), and the replacement of dead corals will undoubtedly take decades while many acute and chronic stressors operate on smaller temporal scales[56]. This is particularly relevant given that corals weakened by SCTLD are likely to be more

susceptible to subsequent disease outbreaks and to other threats like bleaching.

One key question for the coming years is whether populations of highly afflicted species will be able to recover and sustain key geo-ecological functions. To date, we have little evidence in this regard. Empirical observations during our surveys have shown that small (<5 cm) *Meandrina*, *Diploria*, or *Pseudodiploria* corals have apparently remained unaffected by SCTLD, even in sites that underwent severe losses in adult populations (Fig. S6). Furthermore, recent evidence shows that coral colonies with evident lesions associated with SCTLD can spawn and produce viable gametes[57]. However, the replacement of dead corals through larval recruitment or the growth of small juveniles will take several years given low larval survival and the slow growth rates of most species[58,59]. In addition, coral recruitment (i.e., the successful settlement of coral larvae) and subsequent survival will largely depend on suitable ecological conditions, such as low densities of harmful fleshy macroalgae[60]. Unfortunately, in our study region and elsewhere in the Caribbean, there is extensive evidence indicating that macroalgae cover is progressively becoming a dominant component of benthic reef communities[61,62]. It is likely that macroalgae will rapidly overtake the free space left by recently deceased corals (e.g.,[27]), hindering coral recovery by impeding the settlement of new recruits and reducing the likelihood of colonies being able to either slow or halt disease progression by recolonizing their own structures or neighboring substrates (e.g.,[32,63]). Natural processes might therefore be insufficient to restore the severe population losses of many coral species due to the SCTLD outbreak. Rather, it is likely that human interventions in the form of rescuing colonies of vulnerable species, preserving their genetic material, and implementing restoration efforts will be needed to facilitate recovery and prevent the region-wide extinction of some species (e.g.,[28]). We believe, however, that these actions will only succeed if they are accompanied by stringent controls that take into consideration climate change, coastal development, and wastewater treatment to improve local conditions and ecosystem resilience.

## Methods
**Field surveys and SCTLD prevalence.** To assess the spread and impacts of SCTLD, extensive surveys were conducted across the Mexican Caribbean between July 2018 and January 2020 (post-outbreak period). In total, 101 sites were surveyed (82 fore-reefs, 19 back-reefs, and four reef-crests) in depths ranging from 1 to 24 m (Fig. 2; Supplementary Data 1). Thirty-five of these sites were also surveyed in 2016 and 2017 (pre-outbreak period) as part of a separate effort[24,34,64] and were used to investigate the ecological and functional consequences of the SCTLD outbreak. These 35 resampled sites are also distributed across the Mexican Caribbean (Fig. 2) and cover similar habitats and depth gradients (thirty reef sites are fore-reefs and five are back-reefs). For the pre-outbreak period, white plague-type disease prevalence is reported, as there were no reports of SCTLD and a notable overlap between host ranges and the susceptibility to both diseases was present.

All sites were surveyed using the Atlantic and Gulf Rapid Reef Assessment protocol[65]. At each site, coral assemblages were surveyed in $10 \times 1$ m transects that were haphazardly placed within the reef structure at the same depth and reef-zone. For the pre-outbreak period, 1–7 transects (mean = 2.8; SD = 1.4) were evaluated in each site, and for the post-outbreak period between 3–23 transects (mean = 7.1; SD = 3.3) were conducted in each site. The higher number of transects conducted in the post-outbreak period was not expected to artificially increase the relative abundance of coral colonies nor disease prevalence, as the pre-SCTLD effort was already robust enough to capture ecological patterns[65]. Rather, we increased the effort to increase the probability of recording rare species that we knew were highly vulnerable to disease. The following information was recorded for each living coral colony within each transect: species name, colony size (maximum diameter, diameter perpendicular to the maximum diameter, and height), bleaching percentage, partial mortality percentage (new, transition, and old), and the presence of SCTLD or other diseases[65]. For this study, we also recorded colonies with 100% mortality that could be attributed to SCTLD (i.e., recent or transient mortality was still evident; e.g., Fig. 1e). Mortality was deemed to be recent when the superficial structure of the colonies was bare (white coral skeleton) or covered by a thin layer of sediment or filamentous algae, indicating that the soft tissue had died within a time frame of hours to weeks. Only 241 (out of 29,095) post-outbreak colonies were recorded to have been affected by other diseases, none with evidence of rapid disease progression or severe coral mortality. During the surveys, we did

not find evidence of the outbreak of other diseases, therefore, we assumed that coral mortality was produced by SCTLD because this disease has a high rate of lethality and progression[19,26]. Also, to differentiate SCTLD from bleaching, we carefully observed if the colony presented live tissue. When a colony presented signs of bleaching, the remaining tissue had a pale or transparent color that was still visible, which contrasts with what is present with SCTLD, as SCTLD kills the living tissue of the colony. We considered both diseased and recently deceased colonies to prevent underestimating the effects of the disease, as has been done in other similar studies[24,25,27].

We focused on exploring the prevalence and geographical and temporal trends of the most 'highly susceptible species', which were species with more than 10% disease prevalence (considering diseased and recently deceased colonies). These species are: *Colpophyllia natans, Dendrogyra cylindrus, Diploria labyrinthiformis, Dichocoenia stokesii, Eusmilia fastigiata, Isophyllia rigida, Montastraea cavernosa, Meandrina meandrites, M. jacksoni, Mycetophyllia lamarckiana, Orbicella annularis, O. faveolata, O. franksi, Pseudodiploria clivosa, P. strigosa* and *Siderastrea siderea* (Fig. 1b). For this, we used the information from the 101 sites surveyed during the post-outbreak period to calculate SCTLD prevalence for each site and coral species. Although *S. siderea* showed different signs of infection (termed white-blotch syndrome in some studies;[25]), we considered this species to be affected by SCTLD due to the timing of the onset of signs and the disease progression being similar to what we observed with other species. In addition, to generate an overview of the outbreak status of the Mexican Caribbean, we calculated SCTLD prevalence for each site and each period using only highly susceptible species.

### Effect of environmental and anthropogenic covariates on SCTLD prevalence.
We modeled the percentage of afflicted colonies as a function of coral colony density (prior to the impacts of the disease), reef structural complexity, reef zonation, depth, and the degree of exposure to dominant winds. In addition, we evaluated the influence from land-based human activities using the Coastal Development level (World Resources Institute database,[66]) and protection status based on MPA age. These variables were selected based on their importance to coral reef health and the known susceptibility of coral assemblages to disturbance ([34,64], see Table S5 for details), depending on the availability of information for the 101 sites (Supplementary Data 2). Water temperature or thermal stress were not included, as remote sensing data do not capture local variation at the necessary resolution (4 km;[67]). Furthermore, previous studies have shown that temperature is a poor predictor of the spatial variation of coral reef conditions in this ecoregion[68] and that high water temperatures do not affect SCTLD prevalence or virulence[21,24,26,27].

We used a binomial logistic generalized linear mixed model, setting the percentage of afflicted colonies as the response variable (i.e., total number of colonies/number of colonies afflicted) and the aforementioned factors as the predictive variables. All continuous predictive variables were scaled to z-scores with the *scale* function in R. In the model, categorical variables, such as coastal development (low), leeward reefs, and back-reefs, were used as arbitrary references. We included site and transect nested within site as random effects to account for spatial heterogeneity and site-level stochasticity due to the repeated sampling of sites. Moreover, we also included species as a random effect in the regression model to account for species identity and that individuals from the same species may respond to SCTLD more similarly when compared with individuals of other species. Furthermore, we included the sampling size area as a weighting factor to account for the potential effect of effort among sites into our analysis. Statistical analyses were carried out using a 95% confidence interval ($\alpha = 0.05$), and model assumptions were validated with residual plots. Regression models were constructed using the *glmer* function within the *lme4* package[69] in R Version 3.6.1[70].

### Coral morpho-functional groups and community shifts.
The functional diversity of coral communities was estimated using six different traits: skeletal density, growth rate, rugosity index, colony size, reproduction strategy, and corallite width (Table S6). Some or all of these traits have been previously used in other studies to represent the functioning of reef-building corals in a multidimensional space[33,71]. The information on species-level traits was obtained from different sources that provide comprehensive details for each selected trait ([32,72,73]; Table S7). To better compare traits given the contribution disparities between species (see[74,75]), the traits were categorized into numerical groups (1–5). Hierarchical clustering was performed to identify groups in the data set and estimate trait similarity. We grouped the reef corals into morpho-functional groups using a Gower dissimilarity matrix ('vegan' package in R;[76]) and average-linkage hierarchical clustering, which calculates the average distance between clusters before merging ('cluster' package in R;[77]). Groups were defined using 65% dissimilarity because it was the most evident grouping and provided a concise number of groups (Fig. 1).

We measured the contribution of each species to compositional similarity within periods and dissimilarity between periods with a similarity percentage (SIMPER;[78]) analysis. SIMPER identifies the species that are most responsible for the observed patterns (e.g., the species that typify each factor level and those that contribute the most to dissimilarity between levels) by disaggregating the Bray-Curtis similarities between samples. The more abundant a species is within a group,

the more it contributes to intra-group similarity; species with consistently high contributions to the dissimilarity between groups are good discriminating species[79].

To explore temporal changes in coral composition and the traits of those assemblages, a principal component analysis (PCA) was performed using the density (ind/10 $m^2$) of the colonies of each species in each of the 35 reef sites with pre- and post-outbreak information. We evaluated changes in coral-assemblage composition at species and family levels. Changes in functional trait assemblages were assessed through a community-weighted means (CWM) analysis that calculates the relative contribution of a given trait to the coral assemblage, which largely determines ecosystem processes[80,81]. Changes in the functional diversity of coral assemblages were evaluated with functional richness, which represents the volume of the convex hull covering all species in the functional space, and functional evenness, which measures the regularity of the abundance distribution in the functional space[82]. These indices have been widely used to account for functional diversity changes in coral communities[83,84]. Statistical differences between periods were tested with a paired Welch's t-test. All statistical and functional diversity analyses were performed in R[70] using the 'FD' package[85].

### Coral community calcification.
To estimate the effects of SCTLD on the physical functionality of coral reefs, we calculated the potential calcification (i.e., kg CaCO$_3$ m$^{-2}$ yr$^{-1}$) of the coral assemblages for each period using the sum of the calcification rate of each colony proportional to the sampled reef area ($m^2$). We used a reef-level estimation to calculate mean coral community calcification for each period. For this, we estimated the calcification rate of each colony within each study site considering the size, mean annual growth rate (cm yr$^{-1}$), mean skeletal density (g cm$^{-3}$), and morphological growth of each species following the methodology described by Gonzalez-Barrios et al.[86] (see also[31]). For each period, the calcification rate was estimated as the annual volume of calcium carbonate accumulated by the living tissue of the colony for each time period using the information from the 35 resampled sites.

### Statistics and reproducibility.
Raw data are accessible in Supplementary Data 1 and 2. The methods for statistical analysis and sizes of the samples (defined as n) are given in the respective sections of results and methods. Statistical analyses were conducted in R Version 3.6.173 using the cited packages, and reproducibility can be achieved using the parameters reported in the Methods.

### Reporting summary.
Further information on research design is available in the Nature Research Reporting Summary linked to this article.

## Data availability
All data are available in the main text or in Supplementary Data 1 and 2.

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

## Acknowledgements
This study was supported by the Mexican Council of Science and Technology (CON-ACyT, grant number: FORDECYT-PRONACES/425888/2020), the Comisión Nacional de Áreas Naturales Protegidas of Mexico (grant number: PROREST/CER/56/2019), the Universidad Nacional Autónoma de México (UNAM; UNAM-DGAPA-PAPIIT grant number: IN-205019), and a Royal Society Newton Advanced Fellowship (grant number: NA150360). We are grateful three anonymous referees for their comments and suggestions that greatly improved our analysis and the manuscript. We also thank Andrea Lievana for editing the manuscript.

## Author contributions
Conceptualization, Funding acquisition, Project administration, and Writing original draft: L.A.-F.; Methodology and Investigation: L.A.-F., F.J.G.-B., E.P.-C., A.M.-H., N.E.-S.; Data curation: E.P.-C., N.E.-S.; Formal analysis: L.A.-F., F.J.G.-B., A.M.-H., N.E.-S.; Visualization: L.A.-F., F.J.G.-B., N.E.-S.; Writing – review & editing: F.J.G.-B., E.P.-C., A.M.-H., N.E.-S.

## Competing interests
The authors declare no competing interests.
