## [Peer Review File · Communications Biology]

Reviewers' comments:

Reviewer #1 (Remarks to the Author):

Review of Alvarez-Filip et al. "An emerging coral disease outbreak decimated Caribbean coral populations and reshaped reef functionality."

Major comments

This is a well-written and interesting manuscript that details the consequences of the arrival SCTL on the northern Mesoamerican reef. The manuscript's strengths are its quality of writing, its use of complementary analytical approaches, and its tables and figures. I believe the manuscript's main weakness is how disease prevalence was modeled. The authors used linear regression to model disease prevalence (% of afflicted colonies) as a function of various environmental, species-, site-specific predictor variables. The notion of using regression to do this is fine, of course, but using a linear regression seems like an odd choice to me, especially when the authors were already fitting a generalized linear model (albeit one with normal errors and an identity link function). I was also surprised to read that the residual assessment indicated no problems with model goodness-of-fit, especially because no transformation was done on the response variable. I understand that interpreting residual plots is not an exact science, nor is interpreting results from standard tests of normality and heteroscedasticity; however, as a reviewer, I'd prefer to see residual plots from the regression models, and I do think that results from other relevant tests of normality (e.g., Shapiro-Wilk or KS tests) and heteroscedasticity (e.g., Breusch-Pagan test) should be reported in the manuscript, because together they all provide a comprehensive assessment of model fit. More importantly, based on the nature of the data and the study's objectives, it seems to me a logistic regression likely would have been a much better choice here. The authors have all of the required information necessary for creating a binomial response variable (number_of_infected_corals / total_number_of_corals) so fitting a binomial logistic regression would require very minor modifications to their existing R code e.g., change:

```
glm(percent ~ cov1 + cov2 + cov3, data = yourData)
```

to

```
glm(number_afflicted/number_total ~ cov1 + cov2 + cov3, weights = number_total, family = binomial, data = yourData)
```

A binomial process appears to be a much more natural choice for modeling the true underlying data-generating process here (i.e., errors associated with proportion data are generally not normally distributed). I also think that a mixed/random-effects model would have been more appropriate here to account for potential spatiotemporal clustering of observations e.g., via the inclusion of a random intercept that accounts for clustering of observations from multiple transects collected from the same location/time.

The lme4 and glmmTMB packages in R are very useful for fitting such a model, e.g.,

```
glmmTMB(number_afflicted/number_total ~ cov1 + cov2 + cov3 + (1|Site), weights = number_total, family = binomial, data = yourData),
```

 where "Site" is a grouping variable for a random intercept representing a collection of transects sampled from the same location.

Additionally, due to the sheer number of taxa in this study, I think it's fine to not explicitly include species as a fixed effect in the regression model, especially given historical densities were included as a predictor variable; however, I don't think that means you should ignore species identity entirely. I suggest the inclusion of a species-level random effect in the regression model, as I think it's reasonable to account for the fact that individuals from the same species may respond to SCTL more similarly than do to other species.

```
glmmTMB(number_afflicted/number_total ~ cov1 + cov2 + cov3 + (1|Site) + (1|Species), weights = number_total, family = binomial, data = yourData)
```

It's possible that after implementing these suggestions, the authors' main conclusions will remain unchanged, and if all aspects of goodness-of-fit are fine for the linear regression, then everything is fine as-is. If the findings differ, however, or if additional goodness-of-fit assessments indicate the linear regression models do not fit as well as previously thought, then I suggest switching to a mixed effects logistic regression model. I will add that I attempted to re-create the regression analysis using the data submitted as supplemental material, but I found that (a) the general data preparation for regression modeling was not very clearly explained in the Methods, and (b) many of the covariates used in the regression modeling were not present in the Excel spreadsheet. All told, I think it's preferable for authors to supply raw data (as it appears has been done here) as

well as the actual data used for each analysis.

Despite my concerns about the linear regression modeling, the multivariate statistical approaches employed by the authors seem appropriate and well-explained. The paper's tables and figures are also formatted well and informative, especially Figure 1.

Overall, I think this is an important and well-written paper that has the potential to contribute valuable information to the growing body of literature surrounding the short- and long-term consequences of SCTLD on Caribbean coral reefs. As such, I would be very interested to see a revised version of the manuscript that at the very least addresses my comments about the linear regression modeling.

Minor line-by-line comments

Line 32: Stating "will once again become conspicuous..." implies readers should know this already, but many may not. Suggest rephrasing or expanding on this point.

Line 40: I understand what's meant by "the consequences exceed the species level..." but perhaps rephrase to make this clearer to readers.

Line 44: Very interesting to think about the consequences with respect to monitoring sea level rise.

Line 54: Suggest changing to "The disease spread..."

Line 116: When summarizing results like these, I find it often helps to remind readers specifically which analysis is being discussed. For example, instead of saying "The affectation of the SCTLD outbreak...", perhaps simply say "Disease prevalence..." here? Also, see major comments regarding the linear regression analysis.

Lines 117-118: Interesting. This is similar to the Dry Tortugas region in the Florida Keys. Unfortunately, SCTLD eventually arrived there in Summer 2021.

Line 121: Including this particular data set as supplemental material would be useful.

Line 132: I know there is more informing this statement than just the linear regression modeling, but as it stands, I'm not convinced this can be stated given my concerns about those models.

Line 338: The fact that multiple transects were surveyed per site is the reason I think a random site effect is probably warranted in the regression models.

Line 346: As stated earlier, I would have appreciated the inclusion of this data set, as I had a difficult time replicating it from the data that were provided as supplemental material.

Line 361: Please see major comments regarding the regression modeling.

Reviewer #2 (Remarks to the Author):

An emerging coral disease outbreak decimated Caribbean coral populations and reshaped reef functionality by Lorenzo Alvarez-Filip^{1*}, F. Javier González-Barrios¹, Esmeralda Pérez-Cervantes¹, Ana Molina- Hernandez¹, Nuria Estrada-Saldívar¹ Communications Biology manuscript COMMSBIO-21-3061-T

Overview: Authors examine coral disease survey data from the Quintana Roo region (Yucatan) to document the spread of SCTLD and its effects on community structure, specifically the influence of coral traits on disease and effects of disease on calcification. Some of this replicates previous work (3,4) albeit in a more comprehensive manner. Upon reading this, I was left with the questions of 1) Why was this study done? 2) How substantially different is it from what has been previously presented on the topic (3,4), and 3) How does this help us address root causes of SCTLD or its management in the region? The methods are suspect in that it is difficult to disentangle causes of temporal trends in disease (is it effort or true increases). It is unclear how knowing traits of corals affected related to SCTLD. The discussion does not seem to relate back to results in a convincing way. In summary, the paper could use focus and a more coherent synthesis of data in relation to how it substantially adds to our knowledge of SCTLD in the Caribbean.

Title: Consider including "stony coral tissue loss disease" in the title as that seems to be focus of your paper.

Line 58: There is actually now clear evidence at the microscopic level that SCTLD is a breakdown in host/symbiont relationship (1) likely caused by a virus infection killing the symbionts (2). Bacteria actually do not play that much of a role.

Line 118: This is assuming Banco Chinchorro has the same population composition of corals at risk as other sites affected with SCTL. Is that the case? Could it be that BC has low prevalence because there is 1) either lower coral cover or 2) vastly different species composition (e.g. mostly *Acropora*)?. Hard to know looking at Figure 2.

Line 123: Clearly contagiousness has to do with more than just transport of pathogen. Host immune response must be playing a role, because *Acropora* are unaffected but presumably exposed.

Lines 125-130: This is all highly dubious. Suggest delete.

Lines 132-143: If your design was not really structured to address environmental cofactors, then suggest delete all those analyses to simplify.

Line 172: Doubtful-see above.

Lines 183-184: Precisely. Effort. It is really difficult to know here whether the changes you are seeing are a result of true increase in disease or increase in effort. There needs to be a way to control for that in your analyses.

Line 185: Inability to accumulate CaCO_3 or death and increased bioerosion? Inability to accumulate CaCO_3 reads as if corals are losing ability to calcify, but I don't think SCTL does that. I think you mean that increased mortality is leading to more bioerosion and net loss of carbonate budgets.

Line 264: Do the structures persist after death? What kind of bioerosion occurs in dead corals in the area?

Line 268: This appears to contradict what you said in previous paragraph.

Lines 275-277: "It may be necessary to focus on replenishing and favoring the recovery of coral communities while improving our understanding of how to control and modulate the destructive forces operating within coral reefs." How exactly are your findings going to help towards those goals?

Lines 280-281: One could argue that the effects of SCTL in the Caribbean are already well known. See all the papers on the topic that have appeared since 2014.

Lines 279-295: How exactly is all this relevant to your data?

Line 329: Might also want to cite here this paper (3) for methods that presents very similar data on distribution of SCTL in MX.

Line 332: We have here a problem of inadequate case definitions. Grossly, WP and SCTL are very similar (acute to subacute tissue loss) albeit affecting different species. Bottom line, both are unexplained tissue loss. SCTL has a better case definition (both histopathology and TEM). My point is that all you really know is that unexplained tissue loss was less common before than currently, but they could be the same disease or etiology. You don't know absent additional laboratory examinations.

Line 338: Although I do not doubt that SCTL is running rampant in Yucatan, the "increase in effort" makes one question whether increased disease over time is partly an artifact of methods (the more you look the more likely you are to find). Any way to correct for that?

Line 346: Might be good to list those species somewhere.

Line 377: Need citation for R.

Figure 1. It would be useful to have a scale colorbar indicating what range of prevalence values are indicated by the shading of the squares in the dendrogram (e.g. why are some squares darker than others?). If deeper shading indicates higher prevalence, then I don't understand because SCTL D hardly affects *Acropora*. Also, why is this information important? Are there particular traits that are making corals more susceptible to SCTL D? If so, how does knowing this help us address the disease? I note that *Acropora* have many overlapping traits with highly susceptible species. Just how useful is this figure?

Figure 2. Insets in A and B need labels on the x axis. Unclear what this is showing other than lower density of corals from time period 1 to 2.

Figure 3. Right panels...were these differences statistically different?

Figure 4. Unclear where data is coming from to make this figure (specifically the numbers on the Y axis). Couldn't *Orbicella* be considered a massive coral? This figure seems made up to make a nice story but not sure how it is generated. What is defined as "other syndromes". What defines frequent bleaching?

References

- 1) Landsberg, J. H., Y. Kiryu, E. C. Peters, P. W. Wilson, N. Perry, Y. Waters, K. E. Maxwell, L. K. Huebner, and T. M. Work. 2020. Stony coral tissue loss disease in Florida is associated with disruption of host-zooxanthellae physiology. *Frontiers in Marine Science* 7:1090
- 2) Work, T. M., T. M. Weatherby, J. H. Landsberg, Y. Kiryu, S. M. Cook, and E. C. Peters. 2021. Viral-Like Particles Are Associated With Endosymbiont Pathology in Florida Corals Affected by Stony Coral Tissue Loss Disease. *Frontiers in Marine Science* 8.
- 3) Alvarez-Filip, L., N. Estrada-Saldívar, E. Pérez-Cervantes, A. Molina-Hernández, and F. J. González-Barrios. 2019. A rapid spread of the stony coral tissue loss disease outbreak in the Mexican Caribbean. *PeerJ* 7:e8069-e8069.
- 4) Estrada-Saldívar, N., B. A. Quiroga-García, E. Pérez-Cervantes, O. O. Rivera-Garibay, and L. Alvarez-Filip. 2021. Effects of the Stony Coral Tissue Loss Disease Outbreak on Coral Communities and the Benthic Composition of Cozumel Reefs. *Frontiers in Marine Science* 8.

Reviewer #3 (Remarks to the Author):

The authors described multiple findings for their impressive field monitoring of Caribbean reefs before and after the start of the SCTL D outbreak. While depressing to hear, the information summarized in this manuscript is invaluable to the field. I very much appreciated the authors pointing out artifacts in their data and added explanations. This was a clear, well-written document which made it easy to evaluate. The conclusions were well justified and their large sample size was very much appreciated. I would gladly support the acceptance of this manuscript.

I only have a few small suggestions:

- 1) Lines 140 - 143 Maybe comment how this could relate to what was published in Aeby et al. 2021 in *Frontiers in Marine Science*. I found this finding to be reminiscent to what I read there.
- 2) Figure 2 - it's unclear what the different between the circles and triangles are, can this be added to the key?
- 3) I suggest there could be descriptive subheadings added to the Results/Discussion sections. That would improve readability in my opinion.

Editor's comments

We therefore invite you to revise and resubmit your manuscript, taking into account the points raised. In particular, we ask that you better delineate the advance of this work over the existing literature, and address all of Reviewer 1's statistical concerns, as well as Reviewer 2's concerns regarding controlling for research effort. Please highlight all changes in the manuscript text file.

R: Thank you for this comment. This study's main objective was to estimate the consequences of coral die-off on the **functional integrity** of reefs affected by SCTLD. This contrasts with our previous reports (e.g., Alvarez-Filip et al 2019, Estrada-Saldivar et al 2021) and other ecological, SCTLD studies that largely have focused on describing species vulnerability and changes in community composition during or after SCTLD outbreak. We have expanded the description of the aims of the study in the last paragraph of the Introduction.

The manuscript has been carefully revised following all comments provided by the referees. In particular, we have followed the statistical recommendations of Reviewer 1 and explained how we controlled for the research effort. A detailed description of how we have amended the manuscript is provided below.

Reviewers' comments:

Reviewer #1 (Remarks to the Author):

Review of Alvarez-Filip et al. "An emerging coral disease outbreak decimated Caribbean coral populations and reshaped reef functionality."

Major comments

This is a well-written and interesting manuscript that details the consequences of the arrival SCTLD on the northern Mesoamerican reef. The manuscript's strengths are its quality of writing, its use of complementary analytical approaches, and its tables and figures. I believe the manuscript's main weakness is how disease prevalence was modeled. The authors used linear regression to model disease prevalence (% of afflicted colonies) as a function of

various environmental, species-, site-specific predictor variables . The notion of using regression to do this is fine, of course, but using a linear regression seems like an odd choice to me, especially when the authors were already fitting a generalized linear model (albeit one with normal errors and an identity link function). I was also surprised to read that the residual assessment indicated no problems with model goodness-of-fit, especially because no transformation was done on the response variable. I understand that interpreting residual plots is not an exact science, nor is interpreting results from standard tests of normality and heteroscedasticity; however, as a reviewer, I'd prefer to see residual plots from the regression models, and I do think that results from other relevant tests of normality (e.g., Shapiro-Wilk or KS tests) and heteroscedasticity (e.g., Breusch-Pagan test) should be reported in the manuscript, because together they all provide a comprehensive assessment of model fit. More importantly, based on the nature of the data and the study's objectives, it seems to me a logistic regression likely would have been a much better choice here. The authors have all of the required information necessary for creating a binomial response variable ($\text{number_of_infected_corals} / \text{total_number_of_corals}$) so fitting a binomial logistic regression would require very minor modifications to their existing R code e.g., change:

```
glm(percent ~ cov1 + cov2 + cov3, data = yourData)
```

to

```
glm(number_afflicted/number_total ~ cov1 + cov2 + cov3, weights = number_total, family = binomial, data = yourData)
```

A binomial process appears to be a much more natural choice for modeling the true underlying data-generating process here (i.e., errors associated with proportion data are generally not normally distributed). I also think that a mixed/random-effects model would have been more appropriate here to account for potential spatiotemporal clustering of observations e.g., via the inclusion of a random intercept that accounts for clustering of observations from multiple transects collected from the same location/time.

The lme4 and glmmTMB packages in R are very useful for fitting such a model, e.g., `glmmTMB(number_afflicted/number_total ~ cov1 + cov2 + cov3 + (1|Site), weights = number_total, family = binomial, data = yourData)`, where "Site" is a grouping variable for a random intercept representing a collection of transects sampled from the same location.

Additionally, due to the sheer number of taxa in this study, I think it's fine to not explicitly include species as a fixed effect in the regression model, especially given historical densities were included as a predictor variable; however, I don't think that means you should ignore

species identity entirely. I suggest the inclusion of a species-level random effect in the regression model, as I think it's reasonable to account for the fact that individuals from the same species may respond to SCTL D more similarly than do to other species.

```
glmmTMB(number_afflicted/number_total ~ cov1 + cov2 + cov3 + (1|Site) + (1|Species),  
weights = number_total, family = binomial, data = yourData)
```

It's possible that after implementing these suggestions, the authors' main conclusions will remain unchanged, and if all aspects of goodness-of-fit are fine for the linear regression, then everything is fine as-is. If the findings differ, however, or if additional goodness-of-fit assessments indicate the linear regression models do not fit as well as previously thought, then I suggest switching to a mixed effects logistic regression model. I will add that I attempted to re-create the regression analysis using the data submitted as supplemental material, but I found that (a) the general data preparation for regression modeling was not very clearly explained in the Methods, and (b) many of the covariates used in the regression modeling were not present in the Excel spreadsheet. All told, I think it's preferable for authors to supply raw data (as it appears has been done here) as well as the actual data used for each analysis.

R: We sincerely appreciate the detailed explanation and recommendations made to improve our models of disease prevalence. We agree with your concerns and have followed your advice of implementing a GLMM model using a binomial response variable, including 'Site' and 'Species' as Random effects, and the surveyed area (reflecting the sampling effort) as a Weighting variable. We believe this is a more robust approach and have updated the description of the approximation in the Methods section (Lines 413-425):

"We used a binomial logistic generalized linear mixed model, setting the percentage of afflicted colonies as the response variable (i.e., total number of colonies/number of colonies afflicted) and the aforementioned factors as the predictive variables. All continuous predictive variables were scaled to z-scores with the scale function in R. In the model, categorical variables, such as coastal development (low), leeward reefs, and back-reefs, were used as arbitrary references. We included site and transect nested within site as random effects to account for spatial heterogeneity and site-level stochasticity due to the repeated sampling of sites. Moreover, we also included species as a random effect in the regression model to account for species identity and that individuals from the same species may respond to SCTL D more similarly when compared with individuals of other species. Furthermore, we included the sampling size area as a weighting factor to account for the

potential effect of effort among sites into our analysis. Statistical analyses were carried out using a 95% confidence interval ($\alpha = 0.05$), and model assumptions were validated with residual plots. Regression models were constructed using the `glmer` function within the `lme4` package⁷⁴ in R Version 3.6.1⁷⁵.

The results of the model were broadly consistent with our previous results; however, we have made the appropriate changes in the text (Lines 126-156):

“Disease prevalence (considering both diseased and dead colonies) was consistent across the geography. The only region not afflicted by the disease outbreak (at least until the last survey in December 2021) was Banco Chinchorro (Fig. 2). This is an isolated offshore bank with restricted access that is separated from the mainland by a deep-water channel in which the strong northward Yucatan Current likely acts as a physical barrier to biological connectivity and land-based perturbations. Susceptible species to SCTL D are abundant in Banco Chinchorro (Fig. 2b), thus the absence of the disease is not due to the lack of potential host species.

Across all surveyed sites, disease prevalence in highly susceptible species showed no statistical differences with regard to depth, reef zone, structural complexity, or coral density (Table S1; Fig. S1), suggesting that disease spread is primarily controlled by the capacity of the pathogen(s) to be transported in the water column within and between reef sites. However, we found a strong effect of the threat of coastal development on disease prevalence, indicating that sites close to developed areas were considerably more affected than those in isolated regions (Table S1; Fig. S2a). In addition, wind exposure (i.e., windward areas) and the age of marine protected areas (MPAs) were also related to higher disease prevalence (Table S1; Fig. S2a). The effects of coastal development and exposure might be partially explained by the fact that reefs located on the mainland (all windward sites) are influenced by freshwater inflow from combined runoff and submarine groundwater discharges. This, in addition to low water recirculation typical of reef lagoons in these fringing reefs, increases the influx and retention of sediments, nutrients, and pollutants³⁴, which might serve as vectors or influence disease progression (e.g.,^{15,16,35,36}).

The observed significant effects of coastal development, protection, and wind exposure were driven by the lack of SCTLD in the reef sites of Banco Chinchorro. When the reefs of this offshore-bank were removed from the analyses, no significant effects were observed for any of the covariates (Table S2; Fig. S2b), despite the reefs of the leeward coast of Cozumel remaining in the analysis and in spite of many reef sites of the central Mexican Caribbean being found in areas with small human populations and being officially protected since the 1980s (Fig. 2). Although isolation might provide some degree of protection, when the disease reaches a site, coral mortality and the disappearance of key reef-building species will most likely occur regardless of local-scale environmental conditions given SCTLD contagiousness and the high mortality rate after infection^{19,24,31,37}.

Please also note that the dataframe used for the analysis has now been uploaded as supporting material.

Despite my concerns about the linear regression modeling, the multivariate statistical approaches employed by the authors seem appropriate and well-explained. The paper's tables and figures are also formatted well and informative, especially Figure 1.

Overall, I think this is an important and well-written paper that has the potential to contribute valuable information to the growing body of literature surrounding the short- and long-term consequences of SCTLD on Caribbean coral reefs. As such, I would be very interested to see a revised version of the manuscript that at the very least addresses my comments about the linear regression modeling.

R: We thank your positive feedback on our manuscript. We hope you will consider that the model is now suitable for publication.

Minor line-by-line comments

Line 32: Stating "will once again become conspicuous..." implies readers should know this already, but many may not. Suggest rephrasing or expanding on this point.

R: Changed to:

"This emergent disease is likely to become the most lethal disturbance ever recorded in the Caribbean, and it will likely result in the onset of a new functional regime where key reef-

building and complex branching acroporids, an apparently unaffected genus but one that underwent severe population declines decades ago, will once again become conspicuous structural features in reef systems with yet even lower levels of physical functionality.”

Line 40: I understand what’s meant by “the consequences exceed the species level...” but perhaps rephrase to make this clearer to readers.

R: Reworded to:

“When these events affect foundation species, population losses result in changes in the local environment, upon which a variety of other species depend, and alter the structure and functioning of the entire ecosystem^{4,5}.”

Line 44: Very interesting to think about the consequences with respect to monitoring sea level rise.

R: We agree. Please note that in Lines 230-250, we discuss on the implications of our findings in this context.

Line 54: Suggest changing to “The disease spread...”

R: Changed.

Line 116: When summarizing results like these, I find it often helps to remind readers specifically which analysis is being discussed. For example, instead of saying “The affectation of the SCTLD outbreak...”, perhaps simply say “Disease prevalence...” here? Also, see major comments regarding the linear regression analysis.

R: Changed as suggested.

Lines 117-118: Interesting. This is similar to the Dry Tortugas region in the Florida Keys. Unfortunately, SCTLD eventually arrived there in Summer 2021.

R: Thank you. Last December we had the opportunity to visit several of the sites included in this study and found no evidence of SCTLD. We have updated Line 128 with this information.

Line 121: Including this particular data set as supplemental material would be useful.

R: The data tables have been uploaded to the system with this submission.

Line 132: I know there is more informing this statement than just the linear regression modeling, but as it stands, I'm not convinced this can be stated given my concerns about those models.

R: We have reworded this section. Please see our response to your major comment.

Line 338: The fact that multiple transects were surveyed per site is the reason I think a random site effect is probably warranted in the regression models.

R: We have included site as a random effect to account for the potential spatial clustering of observations.

Line 346: As stated earlier, I would have appreciated the inclusion of this data set, as I had a difficult time replicating it from the data that were provided as supplemental material.

R: The data tables have been uploaded to the system with this submission.

Line 361: Please see major comments regarding the regression modeling.

R: This has been modified following your first major comment.

Reviewer #2 (Remarks to the Author):

An emerging coral disease outbreak decimated Caribbean coral populations and reshaped reef functionality by Lorenzo Alvarez-Filip^{1*}, F. Javier González-Barrios¹, Esmeralda Pérez-Cervantes¹, Ana Molina-Hernández¹, Nuria Estrada-Saldívar¹ Communications Biology manuscript COMMSBIO-21-3061-T

Overview: Authors examine coral disease survey data from the Quintana Roo region (Yucatan) to document the spread of SCTLD and its effects on community structure, specifically the influence of coral traits on disease and effects of disease on calcification. Some of this replicates previous work (3,4) albeit in a more comprehensive manner. Upon reading this, I was left with the questions of 1) Why was this study done? 2) How substantially different is it from what has been previously presented on the topic (3,4), and 3) How does this help us address root causes of SCTLD or its management in the region? The methods are suspect in that it is difficult to disentangle causes of temporal trends in disease (is it effort or true increases). It is unclear how knowing traits of corals affected related to SCTLD. The discussion does not seem to relate back to results in a convincing way. In summary, the

paper could use focus and a more coherent synthesis of data in relation to how it substantially adds to our knowledge of SCTLD in the Caribbean.

R: Thank you for the constructive criticism. In the comments below, we describe how we have addressed your detailed comments and hope that the text now better delineates the advances of this study over those our previous studies. Briefly, this study aims to estimate the consequences of coral die-off on the **functional integrity** of reefs affected by this disease (please see lines 81-83). This contrasts with our previous reports (e.g., Alvarez-Filip et al 2019, Estrada-Saldivar et al 2021) and other ecological, SCTLD studies that have largely have focused on describing species vulnerability and changes in community composition during or after an SCTLD outbreak. Please note that the Introduction already includes the main findings of our previous studies as well as those from other research groups (Lines 60-67).

Our current manuscript is built upon a solid and rapidly growing area of research that focuses on the roles that species play within the community and how trait diversity determines ecosystem functioning (Figure 1a and related analyses). Previous studies (unrelated to SCTLD but duly refereed in the manuscript) have proven this approach to be very useful when attempting to comprehend the functional consequences of ecological changes or the impacts of catastrophic events, such as those associated with thermal stress (e.g., Hughes et al., 2018 in Nature; Gonzalez-Barrios et al., 2021 in GCB; McWilliam, et al. 2020 in PRSB). More importantly, this approximation allows for direct connections to be made between ecological change and the provision of key ecosystem goods and services (e.g., Perry et al., 2018 in Nature), as we discuss in lines 230-255. In addition, we measured changes in the functional diversity of coral assemblages before and after outbreaks with indices that have been widely used to represent the range of functions performed by species within coral communities (e.g., Denis et al 2017 in Scientific Reports; Teixidó et al 2018 in Nature Communications).

To our knowledge, some of our key findings have not been tested elsewhere, although they may have been previously hypothesized. For example, we provide strong evidence showing that the coral species that suffered the most severe losses share key life-history traits (Figure 1). This resulted in transformations towards more homogenous assemblages, as determined by taxonomic and functional trait data, with a notorious lack of contributions from the most severely afflicted species (Figure 3). This ultimately impacted the functional diversity of coral

assemblages and their potential to accumulate calcium carbonates (Figure 3 and functional diversity analyses in methods).

We identified that the text in Lines 95-109 and Figure 2 might show some overlap with the information presented in Alvarez-Filip et al 2019. In both cases, we present a description and a map of disease prevalence in the Mexican Caribbean. However, the current manuscript is an updated and more comprehensive representation of disease prevalence, given its geographical extent (*i.e.*, number of sites) and that all Cozumel and mainland sites were affected by the disease. In contrast, in Alvarez-Filip et al. (2019) many sites were surveyed before or at the onset of the outbreak (as surveying was based on preliminary and rapid response). We believe Figure 2 and the text in Lines 95-109 are a useful element in this manuscript, as they provide context for the disease prevalence modelling and trait and functional analyses.

Lastly, we would like to point out that we did not aim to identify how the traits of affected species are related to SCTL D. This is a different research question that probably requires a different conceptual and analytical approach. Our study estimates the functional diversity of coral communities using six different traits (*i.e.*, skeletal density, growth rate, rugosity index, colony size, reproduction strategy, and corallite width) and then represents the functioning of reef-building corals in a multidimensional space (see Lines 435-440). As mentioned above, this allowed us to show a clear relationship between specific morpho-functional groups and disease susceptibility (Lines 99-102), which was then shown to impact the trait space and functional potential of the community (Figure 3; Lines 190-206).

The last paragraph of the Introduction has been updated to clarify some of the concerns raised (Lines 78-88), but please also see our responses to the comments below:

*"...Here, we used extensive pre- and post-SCTL D data along a 450-km reef track in the Mexican Caribbean to examine the regional effects of this emerging threat and to identify the potential ecological and environmental covariates that influenced the occurrence and severity of SCTL D. We then sought to answer how the severe population declines of SCTL D-afflicted species would impact functional diversity and coral community calcification in Caribbean coral reefs. We characterize the functional diversity of Caribbean coral species using six different traits known to be important for reef physical functionality (*i.e.*, skeletal*

density, growth rate, rugosity index, colony size, reproduction strategy, and corallite width). We then show that coral mortality was widespread and corals with intimate phylogenetic (i.e., families) and functional trait relationships were disproportionately affected by SCTLD, favoring the domination of coral species that poorly contribute to reef functionality."

Title: Consider including "stony coral tissue loss disease" in the title as that seems to be focus of your paper.

R: Changed to "Stony coral tissue loss disease decimated Caribbean coral populations and reshaped reef functionality"

Line 58: There is actually now clear evidence at the microscopic level that SCTLD is a breakdown in host/symbiont relationship (1) likely caused by a virus infection killing the symbionts (2). Bacteria actually do not play that much of a role.

R: We agree; both suggested references were added, and the text was changed to:

"The sources of transmission and specific causative agents are not yet fully understood, although the disease is clearly transmitted through seawater, bacteria are involved at some level in disease progression, and viruses of the algal symbionts have been reported in pathological studies suggesting a disruption of host-symbiont physiology²¹⁻²³."

Line 118: This is assuming Banco Chinchorro has the same population composition of corals at risk as other sites affected with SCTLD. Is that the case? Could it be that BC has low prevalence because there is 1) either lower coral cover or 2) vastly different species composition (e.g. mostly Acropora)? Hard to know looking at Figure 2.

R: Thank you for this comment.

First, we would like to point out that the size of the symbol (circle or triangle) in Figure 2 indicates the percentage of healthy colonies of highly susceptible species based on the total number of surveyed colonies (including all coral species) at each site (lines 163-165). Thus, this figure shows that susceptible species are as abundant in Chinchorro (Figure 2b) as they were in many of the mainland and Cozumel sites before the outbreak (Figure 2a). The absence of the disease in Chinchorro is therefore not due to the lack of susceptible species in these sites. We have made this clear in Lines 131-132: *"Species susceptible to SCTLD are abundant in Banco Chinchorro (Fig. 2b), thus the absence of the disease is not due to the lack of potential host species."*

Second, following the major comment of Reviewer 1, we have updated and improved the description of the modeling approach. We now explicitly state that the identities of the coral species that were included in the model as random effects (see our response to Reviewer's comment above).

Line 123: Clearly contagiousness has to do with more than just transport of pathogen. Host immune response must be playing a role, because *Acropora* are unaffected but presumably exposed.

R: We agree; we have changed the text to *"...suggesting that disease spread is primarily controlled by the capacity of the pathogen(s) to be transported in the water column within and between reef sites"*

Lines 125-130: This is all highly dubious. Suggest delete.

R: This model was changed following the suggestion of Reviewer 1. Please see our response to the first comment of Reviewer 1.

Lines 132-143: If your design was not really structured to address environmental cofactors, then suggest delete all those analyses to simplify.

R: This model was changed following the suggestion of Reviewer 1. Please see our response to the first comment of Reviewer 1.

Line 172: Doubtful-see above.

R: This line refers to Figure 1, which shows that most of the species susceptible to SCTL share key life-history traits. This result is based on the hierarchical clustering analysis that was performed to identify groups in the data set and estimate trait similarity (see Lines 429-440 in the Methods section). The dendrogram lines in Figure 1a indicate the similarities among coral species. The analysis (and figure) shows that most of the coral species in the second morpho-functional group from the top in Figure 1a are the most severely afflicted, including species from the genera *Diploria*, *Pseudodiploria*, *Colpophyllia*, *Montastraea*, *Meandrina*, and *Dendrogyra*.

Lines 183-184: Precisely. Effort. It is really difficult to know here whether the changes you are seeing are a result of true increase in disease or increase in effort. There needs to be a way to control for that in your analyses.

R: Thank you for pointing this out. First, we would like to emphasize that we decided to increase the effort in the “post-SCTLD” period to ensure the representation of uncommon and rare species that we knew were affected by SCTLD. So, we aimed to increase opportunities to record species like *D. cylindrus* that happen to be highly susceptible to SCTLD. Previous studies have highlighted the necessity to increase effort to capture the representativeness of rare species (Thompson & Withers 2003). Please also note that our functional analysis (Figure 3) showed a contraction of the functional space after the SCTLD outbreak despite the increase in effort. This is the opposite of what one might expect if the sampling effort were to have an effect on the observed increase in the relative abundance of coral species or colonies.

In addition, the effort during both periods is considered standard for detecting spatial and temporal patterns in abundance or cover (e.g., Wilson & Green 2009, Facon et al. 2016 Front. Mar. Sci.), especially considering that low variability is expected, as transects were haphazardly placed within the reef structure at the same depth and reef-zone (e.g., Murdoch & Aronson 1999 Coral Reefs). We did not target diseased corals nor sample multiple habitats during surveys. Thus, it is unlikely that the larger number of transects in the post-outbreak period would have artificially increased the relative abundance of species or disease prevalence in the same habitat type and depth.

We have improved our description of the field methods to clarify our approach in Lines 371-383:

“All sites were surveyed using the Atlantic and Gulf Rapid Reef Assessment protocol⁶⁹. At each site, coral assemblages were surveyed in 10 x 1 m transects that were haphazardly placed within the reef structure at the same depth and reef-zone. For the pre-outbreak period, 1–7 transects (mean = 2.8; SD = 1.4) were evaluated in each site, and for the post-outbreak period between 3–23 transects (mean = 8; SD = 3.71) were conducted in each site. The higher number of transects conducted in the post-outbreak period was not expected to artificially increase the relative abundance of coral colonies nor disease prevalence, as pre-SCTLD effort was already robust enough to capture ecological patterns⁶⁹. Rather, we increased the effort to increase the probability of recording rare species that we

knew were highly vulnerable to disease. The following information was recorded for each coral colony within each transect: species name, colony size (maximum diameter, diameter perpendicular to the maximum diameter, and height), bleaching percentage, mortality percentage (new, transition, and old), and the presence of SCTL D or other diseases⁶⁹. For this study, we also recorded colonies with 100% mortality that could be attributed to SCTL D (i.e., recent or transient mortality was still evident; e.g., Fig. 1e)."

In addition, we also realized that in the description of the functional diversity analysis, we did not make it clear that we used coral density (colonies/10 m²) to control for the different effort between periods. This has been amended in **Lines 450-452**:

"To explore temporal changes in coral composition and the traits of those assemblages, a principal component analysis (PCA) was performed using the density (ind/10 m²) of the colonies of each species in each of the 35 reef sites with pre- and post-outbreak information."

Lastly, please note that we included the effect of sampling effort when modeling the effects of environmental covariates on SCTL D prevalence. Please see our response to the first comment of Reviewer 1. The effect of sample size did not have any evident effect on the observed patterns of disease prevalence.

Line 185: Inability to accumulate CaCO₃ or death and increased bioerosion? Inability to accumulate CaCO₃ reads as if corals are losing ability to calcify, but I don't think SCTL D does that. I think you mean that increased mortality is leading to more bioerosion and net loss of carbonate budgets.

R: Thank you. This line is about the observed reduction in coral community calcification (Fig. 3f) and not the inability of specific coral colonies to accumulate CaCO₃. We have reworded this for clarity:

"Ultimately, increased mortality was reflected in a marked reduction in coral community calcification (regional mean \pm SE; 4.60 ± 0.77 G = Kg CaCO₃ m² yr⁻¹ before the

outbreak to 3.27 ± 0.53 G after the outbreak; $t = -3.005$, $df = 34$, $p = 0.004$; Fig. 3f) that was largely driven by the loss of highly susceptible species (3.04 ± 0.62 G pre-outbreak to 1.91 ± 0.34 G post-outbreak)."

Line 264: Do the structures persist after death? What kind of bioerosion occurs in dead corals in the area?

R: Yes, we highlight that calcium carbonate skeletons remain in place after tissue mortality. We have reworded this line to improve clarity:

"While key processes related to reef construction and the capacity to track sea-level rise come to a halt with coral mortality⁴⁹, the structures provided by the calcium carbonate skeletons of massive and submassive species are likely to remain in place for several years after the living tissues die. Thus, key functional aspects associated with the tridimensionality of the system, such as habitat provision or the modulation of water energy, will remain for a period after the death of the corals⁹."

Please also note the expanded discussion of bioerosion and other destructive processes in the following paragraph (see next comment).

Line 268: This appears to contradict what you said in previous paragraph.

R: Thanks for your observation. We have rephrased the lines to improve clarity and added a more detailed description of bioeroders, as suggested in the previous comment (Lines 294-312):

"In the absence of recovery, the ultimate consequences of coral mortality will thus be modulated by destructive forces like bioerosion or the biogenic dissolution of reef structures⁵⁰. If erosive processes equal or exceed reef carbonate production, reef frameworks may be destroyed faster than they are produced, resulting in a net negative carbonate budget^{51,52}. Denude coral skeletons are particularly vulnerable to higher rates of biologically mediated processes that occur when micro-and macro-organisms colonize and feed on dead coral structures. For example, increases in light availability in recently dead coral colonies may trigger the chemical dissolution of the skeletons by endolithic light-dependent

*microorganisms*⁵³. The increase in epilithic and endolithic algae on dead coral structures also promotes the grazing activity of sea urchins and parrotfishes, organisms that can remove large amounts of calcium carbonate while feeding, whereas internally, dead coral structures are also colonized by macroborers, such as encrusting sponges, polychaetes, and other bioeroders that can significantly reduce the longevity of individual colonies and gradually weaken carbonate skeletons⁵⁴. Furthermore, future scenarios of ocean warming and acidification predict an increase in environmental conditions that favor destructive forces in coral reefs^{55,56}. Our understanding of how the increased availability of substrate for bioeroders will interact with rapid environmental changes remains limited. However, if the ultimate objective is preserving coral reef functioning and services, it may be necessary to focus on replenishing and favoring the recovery of coral communities while improving our understanding of how to control and modulate the destructive forces operating within coral reefs."

Lines 275-277: "It may be necessary to focus on replenishing and favoring the recovery of coral communities while improving our understanding of how to control and modulate the destructive forces operating within coral reefs." How exactly are your findings going to help towards those goals?

R: Thank you for this comment. In this line, we only want to highlight that given that widespread mortality occurred (as described in the previous paragraphs), it is necessary to not only focus on the replenishment coral communities (which is definitively a priority) but also on increasing our understanding of how destructive forces will operate on dead coral structures. Please note that we have expanded the context of this paragraph, following the previous comment.

Lines 280-281: One could argue that the effects of SCTL D in the Caribbean are already well known. See all the papers on the topic that have appeared since 2014.

R: We agree. However, the point that we wanted to make here is that the disease has not yet spread through the entire Caribbean region. We have reworded this (Lines 314-316) to improve clarity:

"The widespread coral die-off associated with SCTL D has affected the populations of many important reef-building species (Fig.1). Although the impacts of this highly virulent disease

are consistent across affected regions^{19,24-27}, the wide-spread consequences of this outbreak are yet to be known for the entire Caribbean.”

Lines 279-295: How exactly is all this relevant to your data?

R: This paragraph describes the possible implications and consequences for populations affected by SCTL. This is directly linked to our results in Figure 1, which we now refer to in the first sentence of the paragraph. We framed the discussion of the potential futures of these populations within the context of sources of stress and the likelihood of recovery, processes that, although not measured in this study, are essential to understanding the future of the affected species by SCTL.

Line 329: Might also want to cite here this paper (3) for methods that presents very similar data on distribution of SCTL in MX.

R: We have added the citation. Please also see our response to your first comment.

Line 332: We have here a problem of inadequate case definitions. Grossly, WP and SCTL are very similar (acute to subacute tissue loss) albeit affecting different species. Bottom line, both are unexplained tissue loss. SCTL has a better case definition (both histopathology and TEM). My point is that all you really know is that unexplained tissue loss was less common before than currently, but they could be the same disease or etiology. You don't know absent additional laboratory examinations.

R: We agree, although it is important to mention that there is strong overlap between host ranges and the susceptibility of both diseases (e.g., Croquer et al 2021 in Front. Mar. Sci). WP and SCTL are indeed similar, thus we considered all types of white plague-type disease prevalence for the pre-outbreak period. Please also note that disease prevalence of white plague-type diseases was practically absent in the pre-outbreak period (see Figure 2). We have reworded the text to:

“For the pre-outbreak period, white plague-type disease prevalence is reported, as there were no reports of SCTL and notable overlap between host ranges and the susceptibility to both diseases was present.”

Line 338: Although I do not doubt that SCTLD is running rampant in Yucatan, the "increase in effort" makes one question whether increased disease over time is partly an artifact of methods (the more you look the more likely you are to find). Any way to correct for that?

R: Please see our reply to your comment about Lines 183-184.

Line 346: Might be good to list those species somewhere.

R: We have added the list in Lines 387-391. Please note the species were already shown in Figure 1.

Line 377: Need citation for R.

R: We added the citation. This paragraph has undergone notable changes, following the major comment of Reviewer 1.

Figure 1. It would be useful to have a scale colorbar indicating what range of prevalence values are indicated by the shading of the squares in the dendrogram (e.g. why are some squares darker than others?). If deeper shading indicates higher prevalence, then I don't understand because SCTLD hardly affects *Acropora*. Also, why is this information important? Are there particular traits that are making corals more susceptible to SCTLD? If so, how does knowing this help us address the disease? I note that *Acropora* have many overlapping traits with highly susceptible species. Just how useful is this figure?

R: Thank you for pointing this out. The figure description was confusing, and no indication of the color intensity (referred in the comment as shading) was provided. We have amended the figure legend as follows:

"Morpho-functional groups of Caribbean corals and their susceptibility to stony coral tissue loss disease (SCTLD). a) Hierarchical clustering dendrogram based on a Gower dissimilarity analysis and heat map representation of functional traits. Numerical values from 1 to 5 correspond to the categories listed in Table S6. The variation in color intensity within each group (light to deep) corresponds to the trait numerical value (given inside each square)..."

Please also note that Figure 1a is based on a hierarchical clustering analysis performed to identify functional groups in the data set and estimate trait similarity (see Lines 433-440 in the Methods section). The dendrogram lines in Figure 1a indicate the similarities among coral species. The figure shows that the two acroporids are highly similar among themselves

but show the greatest dissimilarity with all other coral species. This is because both acroporids consistently show the highest growth rate, skeletal density, colony size, and colony complexity values. This is not seen with any other species (including those that are highly susceptible to SCTLD).

Lastly, we would like to point out that we did not aim to identify how the traits of affected species are related to SCTLD. This is a different research question that would require a different conceptual and analytical approach. Our study estimates the functional diversity of coral communities using six different traits (i.e., skeletal density, growth rate, rugosity index, colony size, reproduction strategy, and corallite width) to represent the functioning of reef-building corals in a multidimensional space (see lines 429-432). As mentioned above, this allowed us to show a clear relationship between specific morpho-functional groups and disease susceptibility (Lines 100-102), which was then shown to impact the trait space and functional potential of the community (Figure 3; Lines 190-206). Please also note that following your first comment, we have expanded our description of the objectives of this study to improve clarity (Lines 78-88).

Figure 2. Insets in A and B need labels on the x axis. Unclear what this is showing other than lower density of corals from time period 1 to 2.

R: Thank you for pointing this out. The labels "pre-outbreak" and "post-outbreak" were added to the insets.

Figure 3. Right panels... were these differences statistically different?

R: Yes, they are statistically different. Please see lines 198-206.

Figure 4. Unclear where data is coming from to make this figure (specifically the numbers on the Y axis). Couldn't *Orbicella* be considered a massive coral? This figure seems made up to make a nice story but not sure how it is generated. What is defined as "other syndromes". What defines frequent bleaching?

R: Thank you for this comment. This figure is a conceptual diagram, which was constructed based on published data. Following this comment, we have reworded the figure's title and defined physical functionality. We now also include Supplementary Table S4, which provides the rationale and sources used to construct the figure.

Orbicella was considered a separated group, as this is one of the main reef-building genera in the Caribbean, and *Orbicella* have dominated shallow-water coral-reef habitats throughout the region since ~600 thousand years ago (Toth et al 2019 and references there in). The term "other syndromes" was replaced with "other coral-disease syndromes" (*sensu* Harvell et al 2007) to refer to other diseases (e.g., white plague disease and yellow band disease) that have been identified as drivers of the population declines of some massive species. We used the term "syndromes" given that the causative agents have not been identified for many of these diseases. Also, we decided to remove the word "frequent" from "Widespread & frequent bleaching events."

Figure legend now reads as:

"Figure 4. Conceptual diagram of the long-term trajectory of the physical functionality of Caribbean reefs based on published temporal trends (Table S4) and the recent impacts of stony coral tissue loss disease (SCTLD). The physical functionality of reefs depends on the abundance (or cover), capacity to accumulate CaCO₃, and structural complexity of each species present in the system³². The stacked plot represents the functional contributions of four coral groups over time. The pie charts illustrate the proportional contributions of each coral group during three different periods. *Acropora* spp. and *Orbicella* spp. contain all the species for each of these genera and are illustrated as a single group, as they are the main reef-building corals in the Caribbean and have dominated shallow-water coral-reef habitats throughout the region in geological times³⁹. The group of massive corals includes important reef framework builders from the *Diploria*, *Pseudodiploria*, *Colpophyllia*, *Montastraea*, and *Dendrogyra* genera (many of which were severely affected by SCTLD and were included in the second morpho-functional group from the top in Fig. 1a). The other group includes all other coral species, which are largely classified as weedy, submassive, or foliose-digitate corals (included in the third and fourth morpho-functional groups from the top in Fig. 1a) for which little evidence of declines exists. The black arrows indicate major sources of coral decline widely recognized in the literature. White-band disease resulted in severe population declines of acroporids¹⁰. The white-pox epidemic has infected many of the remaining colonies of this genus since the

1990s⁴⁵. Other coral-disease syndromes (e.g., white plague and Caribbean yellow band) that mainly affect *Orbicella* and other massive species have increased in frequency and virulence over the last three decades (e.g., ^{7,46}). Coral mortality has also continued to increase in the Caribbean and is associated with warm-water bleaching events and other local-scale anthropogenic impacts^{13,47,48}. The grey-dashed arrows indicate that the source of stress remains, although the effects on widespread coral mortality are unclear.”

References

- 1) Landsberg, J. H., Y. Kiryu, E. C. Peters, P. W. Wilson, N. Perry, Y. Waters, K. E. Maxwell, L. K. Huebner, and T. M. Work. 2020. Stony coral tissue loss disease in Florida is associated with disruption of host-zooxanthellae physiology. *Frontiers in Marine Science* 7:1090
- 2) Work, T. M., T. M. Weatherby, J. H. Landsberg, Y. Kiryu, S. M. Cook, and E. C. Peters. 2021. Viral-Like Particles Are Associated With Endosymbiont Pathology in Florida Corals Affected by Stony Coral Tissue Loss Disease. *Frontiers in Marine Science* 8.
- 3) Alvarez-Filip, L., N. Estrada-Saldívar, E. Pérez-Cervantes, A. Molina-Hernández, and F. J. González-Barrios. 2019. A rapid spread of the stony coral tissue loss disease outbreak in the Mexican Caribbean. *PeerJ* 7:e8069-e8069.
- 4) Estrada-Saldívar, N., B. A. Quiroga-García, E. Pérez-Cervantes, O. O. Rivera-Garibay, and L. Alvarez-Filip. 2021. Effects of the Stony Coral Tissue Loss Disease Outbreak on Coral Communities and the Benthic Composition of Cozumel Reefs. *Frontiers in Marine Science* 8.

Reviewer #3 (Remarks to the Author):

The authors described multiple findings for their impressive field monitoring of Caribbean reefs before and after the start of the SCTL outbreak. While depressing to hear, the information summarized in this manuscript is invaluable to the field. I very much appreciated the authors pointing out artifacts in their data and added explanations. This was a clear, well-written document which made it easy to evaluate. The conclusions were well justified and their large sample size was very much appreciated. I would gladly support the acceptance of this manuscript.

I only have a few small suggestions:

R: We appreciate your positive and constructive assessment and have amended our manuscript following your suggestions and comments.

1) Lines 140 - 143 Maybe comment how this could relate to what was published in Aeby et al. 2021 in Frontiers in Marine Science. I found this finding to be reminiscent to what I read there.

R: Thank you. We have now included this reference. Please also note that the paragraph has been modified following the major comment of Reviewer 1.

2) Figure 2 - it's unclear what the difference between the circles and triangles are, can this be added to the key?X4

R: Thank you for pointing this out. The circles represent the locations of the reefs that were sampled before and after the outbreak. The triangles represent the sites that were only surveyed during or after the outbreak. The information is now provided in the figure legend (Lines 167-168) to make this clear.

3) I suggest there could be descriptive subheadings added to the Results/Discussion sections. That would improve readability in my opinion.

R: Thank you. We have added subheadings as suggested.

Reviewers' comments:

Reviewer #1 (Remarks to the Author):

The authors have put a substantial amount of effort into revising their analysis of disease prevalence, my main concern in the first draft, and although I think it's a better approach than their initial modeling effort, there still appear to be some problems with the models; namely, the use of Area_m2 in the weights argument. Fortunately, these problems don't seem to influence the study's main findings, at least for the 101-site analysis, but I think they still need to be fixed. I'm a little less clear about the 86-site analysis as I cannot recreate the standardized continuous predictors properly because I don't know the original means and SDs of the variables; hence, it's unclear whether the estimates in Table S2 are correct. Regardless, based on the description of the models in the Methods, I believe that both models (101 and 86 site analyses) appear to suffer from the same problem I describe below and demonstrate in Reviewer 1 attachment #1.

Lines 430 - 432

"Furthermore, we included the sampling size area as a weighting factor to account for the potential effect of effort among sites into our analysis."

It seems that sample area was included in the weights argument, but I do not think that is correct. Generally, I don't really see how area would play a role here given the focus is on proportions (survey area would be a bigger deal if you were modeling counts). Instead, you want to use weights = Total because that tells the model how many trials were associated with each proportion i.e., the amount of information (which could be viewed as effort) contributing to calculation of the proportion. The regression model should therefore look something like this, noting that (1|Site/Transect) is equivalent to (1|Site) + (1|Site:Transect):

```
m1 <- glm(Afflicted/Total ~ Cov1 + Cov2 + Cov3 + ... + (1|Site/Transect) + (1|Species), family = binomial, weights = Total, data = yourData)
```

Please see Reviewer 1 attachment #1 for a more detailed explanation.

TABLES AND FIGURES

Tables S1 and S2 would benefit from slight changes to their format and some additional information. Additionally, in the Figure S2 caption the authors state that the estimates are expressed as odds-ratios, but they are on the log-odds scale. I recommend deleting the sentence about odds ratios and keeping them on the scale of log-odds. Keep in mind, however, that I'm not convinced these estimates are correct given the current model, so this figure will likely need to be revised after the models are refit using weights = Total instead of weights = Area_m2. Please more detailed comments in Reviewer 1 attachment #2.

Reviewer #2 (Remarks to the Author):

Stony coral tissue loss disease decimated Caribbean coral populations and 1 reshaped reef functionality by Lorenzo Alvarez-Filip^{1*} F. Javier González-Barrios¹, Esmeralda Pérez-Cervantes¹, Ana Molina-4 Hernandez¹, Nuria Estrada-Saldívar¹ COMMSBIO-21-3061A

Overview: Thanks to the authors for addressing previous comments. The authors certainly have an interesting story and an impressive data set, but they continue to smother what would otherwise be an interesting narrative with un-necessary jargon, figures, and foggy thinking. Their paper would be much more compelling if they condense, simplify, and stick to their central message (effects of SCTLTD and role of functional traits). Often, less is more. I try here to provide additional input to help this along.

Abstract:

Line 26: "Rapid spread" more appropriate than "infection". We are not completely sure SCTLTD is infectious (although it is increasingly looking like it might be).

Lines 32-34: "...acroporids, an apparently unaffected genus but that underwent severe population declines decades ago, will once again become conspicuous structural features in reef systems with yet even lower levels of physical functionality." How do you reconcile this statement with your previous work (1) indicating that loss of Acropora led to loss of diversity? It would seem that resurgence of Acropora would be a good thing no? Perhaps one way around this is to add a statement to effect that whilst Acropora will be once again dominant, their low rate of recruitment will not compensate for loss of existing reefs and will lead to lower functionality.

Introduction

Lines 85-86: Consider adding a citation for those 6 traits.

Lines 87-89: Consider rewording for clarity "Corals in the family Meandrinidae and Mussidae and with low growth rate and spawning reproduction were disproportionately affected by SCTLTD." Might also want to add this to abstract.

Methods

375-377: Consider deleting the white plague material (see below). This distracts from your central message (Functional traits of corals affected by SCTLTD) and does not add substantially to the story. In any case, who is to say that white plague historically was not SCTLTD? We don't know because no one did laboratory diagnostics to figure it out. And instead of panel A in Figure 2, consider a panel showing percent loss of coral cover over time for each site (or those sites for which you have most robust data). That would be a much more interesting data point illustrating demographic effects of SCTLTD in Yucatan. I realize you looked at temporal trends with PCA, but simple before after histograms might show the same thing in a more easily digestible manner. In fact, consider replacing Figure 2a with Figure S1.

387-389: I do not understand the distinction between mortality percentage and presence of SCTLTD and other diseases. SCTLTD as seen in the field is, by definition, death of coral tissues leading to bare or algae covered skeletons (mortality); it is simply unexplained mortality of corals. A definitive diagnosis (as of right now at least) requires lab tests like histology. So, Consider simplifying and having 2 categories: Bleaching, and unexplained tissue loss (graded as new, transition and old); you can even call unexplained tissue loss "SCTLTD" if you want. Tissue loss and bleaching is really all you can see in the field, so it is more honest assessment of situation and gets you away from artificial contortions of trying to differentiate this or that tissue loss disease from SCTLTD (which you cannot do in the field other than obvious causes such as fish bites, COTS predation or anchor damage).

Lines 401-403: See above for a solution to your *Siderastrea* problem.

Results

Line 108-Consider moving Fig. S1 as a main figure and moving the PCA plots as supplementals. Figure S1 nicely illustrates demographic effects of SCTLTD.

Line 131-132: I would think too an important distinction with Banco Chinchorro is absence of human development there, correct (at least compared to Cozumel)? Might want to add that to narrative if true.

Lines 241-243: "However, the resulting wide-spread coral mortality described here was dictated by the vulnerability of species to SCTLTD, and thus caused non-random changes in community structure that further and radically affected the functional integrity of the coral communities." I don't understand how that makes SCTLTD so special. After all, the widespread decline of Acroporids in the late 1970s-80s were dictated by vulnerability of that particular family to unexplained tissue loss. How is that any different than SCTLTD?

Line 249: I am curious as to definition of an opportunistic coral. For instance, could fast growing *Acropora* be called opportunists too?

Lines 262-274: Lots of jargon to unpack here. How do functions serve to "track" increase sea level? Why is low calcification in other regions of the world relevant to your study site? Can you provide a citation to show that calcification rates were already low prior to SCTL in Caribbean? That would be more germane. How does reef construction "track" sea level rise? What does tridimensionality mean? Seems all of this could be condensed to a single sentence as follows: "Loss of corals to SCTL will lead to further bioerosion of coral reef ecosystems making coastal communities more susceptible to sea level rise caused by climate change."

lines 300-318: You touch on similar themes as above. Suggest you condense these 2 paragraphs into a couple of succinct sentences explaining why loss of corals will lead to bioerosion and increased coastal hazards due to climate change." I think that is what you are trying to communicate.

Line 333: I think you mean here allee effect (2).

Figures

Figure 1. Consider deleting C-E. There are now plenty of photos of SCTL in literature, and these do not add substantially to your message.

Figure 2. Delete Panel A. See my comments re. white plague. Also delete violin plots. By promoting Figure S1 as a main figure, you illustrate there very well effects of SCTL on coral demographics. This then simplifies the figure to single panel (B).

Figure 4. Delete. I'm unconvinced that this figure means anything. It is basically a cartoon trying to illustrate that corals have declined over time in the Caribbean and that relative proportions of different corals have changed over time. All of that has been well documented in literature with concrete survey data (Cramer et al., *Sci. Adv.* 2020; 6 : eaax9395). You can cite those studies to make your point without having to resort to a cartoon. Plus, deleting it will simplify your central message.

Supplemental

Table S1 title: "...significant p values." Also, I do not see bolded p values in that table.

References

- 1) Alvarez-Filip, L, NK Dulvy, JA Gill, IM Cote, and AR Watkinson. (2009). Flattening of Caribbean coral reefs: region-wide declines in architectural complexity. *Proceedings of Royal Society B* 276:3019-3025.
- 2) *Journal of Experimental Zoology*. 61 (2): 185-207.

Editor's comments

We therefore invite you to revise and resubmit your manuscript, taking into account the points raised. In particular, we ask that you consider Reviewer 1's suggestions for the treatment of sample area in the disease prevalence models, and address Reviewer 2's comments regarding the manuscript framework and presentation.

R: We thank you and the reviewers for comments on the manuscript. The manuscript has been carefully revised. We have followed the statistical recommendations of Reviewer 1 and amended the text to improve the manuscript's clarity following the comments of Reviewer 2. We, however, decided to keep some figures Reviewer 2 suggested deleting.

Reviewers' comments:

Reviewer #1 (Remarks to the Author):

The authors have put a substantial amount of effort into revising their analysis of disease prevalence, my main concern in the first draft, and although I think it's a better approach than their initial modeling effort, there still appear to be some problems with the models; namely, the use of Area_m2 in the weights argument. Fortunately, these problems don't seem to influence the study's main findings, at least for the 101-site analysis, but I think they still need to be fixed. I'm a little less clear about the 86-site analysis as I cannot recreate the standardized continuous predictors properly because I don't know the original means and SDs of the variables; hence, it's unclear whether the estimates in Table S2 are correct. Regardless, based on the description of the models in the Methods, I believe that both models (101 and 86 site analyses) appear to suffer from the same problem I describe below and demonstrate in Reviewer 1 attachment #1.

Lines 430 - 432

"Furthermore, we included the sampling size area as a weighting factor to account for the potential effect of effort among sites into our analysis."

It seems that sample area was included in the weights argument, but I do not think that is correct. Generally, I don't really see how area would play a role here given the focus is on proportions (survey area would be a bigger deal if you were modeling counts). Instead, you want to use weights = Total because that tells the model how many trials were associated

with each proportion i.e., the amount of information (which could be viewed as effort) contributing to calculation of the proportion. The regression model should therefore look something like this, noting that (1|Site/Transect) is equivalent to (1|Site) + (1|Site:Transect):

```
m1 <- glm(Afflicted/Total ~ Cov1 + Cov2 + Cov3 + ... + (1|Site/Transect) + (1|Species),  
family = binomial, weights = Total, data = yourData)
```

Please see Reviewer 1 attachment #1 for a more detailed explanation.

R: Thank you for the comments. We sincerely appreciate your observations and guided explanations, which allowed us to improve and strengthen our models. As suggested, we now use the total number of colonies ("Total") as a weighting factor (instead of transect area "Area_m2"). All models and outputs have been updated. Also, to facilitate data handling, we have provided the raw data of the continuous predictors (i.e., without scaling) in the supporting material. Please note that "Reef Zone" is a categorical variable.

As noted in your comments, the results are highly consistent, and the conclusions remain largely the same. The only difference with regard to our previous results is that coastal development was significant in the model without Chinchorro (n = 86). Given this change and because we identified some redundancies in the description of these results, we modified the text to improve clarity (Lines 130-183).

Given the change in the models' outputs described above and the fact the statistical approximation has considerably improved, we decided to move the figure showing the prevalence predictors to the main document (now Figure 3). Also, we added two new figures (Fig S2 and Fig S3) to show the detailed plot models of each disease prevalence predictor as supporting information.

TABLES AND FIGURES

Tables S1 and S2 would benefit from slight changes to their format and some additional information. Additionally, in the Figure S2 caption the authors state that the estimates are expressed as odds-ratios, but they are on the log-odds scale. I recommend deleting the sentence about odds ratios and keeping them on the scale of log-odds. Keep in mind, however, that I'm not convinced these estimates are correct given the current model, so this figure will likely need to be revised after the models are refit using weights = Total instead of weights = Area_m2. Please see more detailed comments in Reviewer 1 attachment #2.

R: Thank you. We have updated tables S1 and S2 according to your suggestion.

Reviewer #2 (Remarks to the Author):

Stony coral tissue loss disease decimated Caribbean coral populations and reshaped reef functionality by Lorenzo Alvarez-Filip^{1*} F. Javier González-Barrios¹, Esmeralda Pérez-Cervantes¹, Ana Molina-Hernández¹, Nuria Estrada-Saldivar¹ COMMSBIO-21-3061A

Overview: Thanks to the authors for addressing previous comments. The authors certainly have an interesting story and an impressive data set, but they continue to smother what would otherwise be an interesting narrative with un-necessary jargon, figures, and foggy thinking. Their paper would be much more compelling if they condense, simplify, and stick to their central message (effects of SCTL D and role of functional traits). Often, less is more. I try here to provide additional input to help this along.

R: Thank you for your comments and suggestions, as they helped us to improve the clarity of the manuscript. We, however, have decided to keep some figures in the manuscript that you had suggested deleting. Below, we provide detailed justifications of why we believe this information is essential to our manuscript.

Abstract:

Line 26: "Rapid spread" more appropriate than "infection". We are not completely sure SCTL D is infectious (although it is increasingly looking like it might be).

R: Changed.

Lines 32-34: "...acroporids, an apparently unaffected genus but that underwent severe population declines decades ago, will once again become conspicuous structural features in reef systems with yet even lower levels of physical functionality." How do you reconcile this statement with your previous work (1) indicating that loss of *Acropora* led to loss of diversity? It would seem that resurgence of *Acropora* would be a good thing no? Perhaps one way around this is to add a statement to effect that whilst *Acropora* will be once again dominant, their low rate of recruitment will not compensate for loss of existing reefs and will lead to lower functionality.

R: Thank you for the comment. Here we do not mean to say that *Acropora* is increasing. We want to make the point that the increased contribution of acroporids is an artefact of the

drastic reductions in the relative contributions of many other species due to SCTL. We have reworded this line in the Abstract to improve clarity:

"... This emergent disease is likely to become the most lethal disturbance ever recorded in the Caribbean, and it will likely result in the onset of a new functional regime where key reef-building and complex branching acroporids, an apparently unaffected genus that underwent severe population declines decades ago and retained low population levels, will once again become conspicuous structural features in reef systems with yet even lower levels of physical functionality."

We would also like to highlight that we expand on this idea in the Discussion, where we have included your valuable point of *Acropora* having low recruitment rates (Lines 262-275).

Lastly, in this context, we would like to take this opportunity to highlight the need for Figure 4. This conceptual figure (based on previous research) represents this very point visually by showing an overall decrease in coral cover and functionality accompanied by changes in the relative contributions of different coral groups. For the 'post-SCTL' period, reefs show very low functionality (and cover) but with disproportionate contributions of acroporids.

Introduction

Lines 85-86: Consider adding a citation for those 6 traits.

R: Thank you. We have added the citation. Please note that more detail on these traits is provided in the Methods and supporting material.

Lines 87-89: Consider rewording for clarity "Corals in the family Meandrinidae and Mussidae and with low growth rate and spawning reproduction were disproportionately affected by SCTL." Might also want to add this to abstract.

R: Thank you. We have included the names of the families in these lines and the Abstract. A detailed description of the trait relationships is given in the first paragraph of the 'Results and Discussion' section (6 lines below), therefore we decided not to repeat this information here.

Methods

375-377: Consider deleting the white plague material (see below). This distracts from your central message (Functional traits of corals affected by SCTLD) and does not add substantially to the story. In any case, who is to say that white plague historically was not SCTLD? We don't know because no one did laboratory diagnostics to figure it out.

R: Thank you for your comments.

Regarding the white plague material, although white plague lacks a detailed microscopic description, previous studies have based their observations in the macroscopic signs that differentiate SCTLD from WP. These include acute multifocal infections on single colonies, the presence of a bleached border separating apparently healthy tissues seems, tissue and mucus sloughing, and rapid tissue mortality (Aeby et al., 2019; Weil et al., 2019; Cróquer and Weil 2021).

However, we agree in that during the 'pre-SCTLD' period, we did not aim to distinguish between white plague and SCTLD. This is why we explicitly use the term "white plague-type diseases." This is also the very reason we consider it important to retain this material (Figure 2a and text). Please note that Figure 2a shows that there were practically no reports of white plague-type diseases before the SCTLD outbreak (all sites colored in blue in Figure 2a). Without this information, it would not be possible to support (1) that the outbreak started in 2018 and (2) that the outbreak is the main cause of the observed changes. Please note that Figure 2a also serves to show the severe declines in the 'post-SCTLD' period of susceptible species (when compared with that is shown in Figure 2b).

References cited in this comment:

Aeby, G. S. et al. Pathogenesis of a Tissue Loss Disease Affecting Multiple Species of Corals Along the Florida Reef Tract. *Frontiers in Marine Science* 6, 1–18 (2019).

Weil, E. et al. Spread of the new coral disease SCTLD into the Caribbean : implications for Puerto Rico. *Reef Encounter* 34, 38–43 (2019).

Cróquer, A., Weil, E. F. & Rogers, C. Similarities and differences between two deadly Caribbean coral diseases: White plague and stony coral tissue loss disease. *Frontiers in Marine Science* 1331 (2021).

And instead of panel A in Figure 2, consider a panel showing percent loss of coral cover over time for each site (or those sites for which you have most robust data). That would be a much more interesting data point illustrating demographic effects of SCTLTD in Yucatan.

R: Thank you for this recommendation. For this study, we are not using coral cover because we do not have coral cover data for most sites. However, we want to direct your attention to the size of symbols in Figures 2a and 2b, as we believe this shows the effect you wanted to see. The size reflects the proportion of susceptible species that underwent severe declines in the 'post-SCTLTD' period (symbols are consistently smaller in Figure 2b compared to those in Figure 2a).

I realize you looked at temporal trends with PCA, but simple before after histograms might show the same thing in a more easily digestible manner. In fact, consider replacing Figure 2a with Figure S1.

R: Thank you for your suggestion. However, Figure S1 and Figure 3a-c do not show the same information.

Figure S1 only shows colony density, lumping all sites for each species before and after the outbreak (i.e., there is no information at the site level or at coral community level). In contrast, the Principal Component Analyses (PCA) show the changes in composition at the site level; the absolute contributions of species, families, or traits to the observed differences; and the 95% confidence intervals that show how different the groups are (i.e., 'pre-outbreak' vs 'post-outbreak'). Ordination plots have been used in similar ecological studies (e.g., Bjorkman et al., 2018; Hughes et al., 2018; González- Barrios et al., 2021) and allow us to represent the drastic transformation towards more homogenous assemblages, as determined by taxonomic and functional trait data, with a notorious lack of contributions from the most severely afflicted species during the post-outbreak period (Fig. 3a-c).

If we were to replace figure 3a-c with Figure S1, we would only be able to say that in general (i.e., regional level), the number of colonies of some species declined after the outbreak, but we would not be able to describe changes in the taxa and trait composition. For this reason, we have decided to keep figure 3a-c in the main text and Figure S1 as supporting material, as we consider it provides complementary information to Figure 1b and Figure 3a-c.

References cited in this comment:

- Bjorkman, A. D. et al (2018). Plant functional trait change across a warming tundra biome. *Nature*, 562(7725), 57-62.
- Hughes, T. P., et al (2018). Global warming transforms coral reef assemblages. *Nature*, 556(7702), 492-496.
- González- Barrios, F. J., et al (2021). Recovery disparity between coral cover and the physical functionality of reefs with impaired coral assemblages. *Global Change Biology*, 27(3), 640-651.

387-389: I do not understand the distinction between mortality percentage and presence of SCTLD and other diseases. SCTLD as seen in the field is, by definition, death of coral tissues leading to bare or algae covered skeletons (mortality); it is simply unexplained mortality of corals. A definitive diagnosis (as of right now at least) requires lab tests like histology. So, Consider simplifying and having 2 categories: Bleaching, and unexplained tissue loss (graded as new, transition and old); you can even call unexplained tissue loss "SCTLD" if you want. Tissue loss and bleaching is really all you can see in the field, so it is more honest assessment of situation and gets you away from artificial contortions of trying to differentiate this or that tissue loss disease from SCTLD (which you cannot do in the field other than obvious causes such as fish bites, COTS predation or anchor damage)

Thank you for your observation.

Mortality percentages were recorded for all living colonies, not just for the ones with SCTLD. We have now made this clear in the text. Regarding diseases, there are macroscopic signs that are used as a diagnostic criteria. With these we could discerned between SCTLD, Yellow Band Disease, and Black Band Disease and their associated resulting mortality rates in living colonies (Raymundo et al., 2008; Bourne et al., 2022).

With regard to total mortality, we considered dead colonies as those affected by SCTLD to reflect the magnitude of its impact. Please consider the following:

1. SCTLD disease can kill a coral colony in a period of days to weeks. If we only considered living corals at the time of the surveys, we would not be able to describe the real magnitude of the outbreak.

2. SCTL D was the only major disturbance to affect coral communities during the time span of our study. This may explain the high coral mortality observed and coincides with the diagnosis of SCTL D given that this disease has high rates of lethality, dispersion, and progression (Precht et al., 2016; Muller et al., 2020).

3. We only considered recently dead colonies as having been afflicted by SCTL D. These were the colonies in which the superficial structures were either bare (white coral skeletons) or covered by thin layers of sediment or filamentous algae, indicating that the soft tissue had died within hours to weeks (Fig. 1e).

4. The inclusion of recently dead corals was consistent with the SCTL D epidemic. The same approximation has also been made by other research groups with the objective of not underestimating of the effects of the disease (e.g., Miller et al., 2016; Neely, 2018; Gintert et al., 2019; Dahlgren et al., 2021).

We have now made it clear in the text that our approximation is consistent with those previous studies, and we have provided additional details on the criteria and rationale used in the Methods section (Lines 392-405):

"... For this study, we also recorded colonies with 100% mortality that could be attributed to SCTL D (i.e., recent or transient mortality was still evident; e.g., Fig. 1e). Mortality was deemed to be recent when the superficial structure of the colonies was bare (white coral skeleton) or covered by a thin layer of sediment or filamentous algae, indicating that the soft tissue had died within a time frame of hours to weeks. Only 241 (out of 29,095) post-outbreak colonies were recorded to have been affected by other diseases, none with evidence of rapid disease progression or severe coral mortality. During the surveys, we did not find evidence of the outbreak of other diseases, therefore, we assumed that coral mortality was produced by SCTL D because this disease has a high rate of lethality and progression^{19,26}. Also, to differentiate SCTL D from bleaching, we carefully observed if the colony presented live tissue. When a colony presented signs of bleaching, the remaining tissue had a pale or transparent color that was still visible, which contrasts with what is present with SCTL D, as SCTL D kills the living tissue of the colony. We considered both diseased and recently deceased colonies to prevent underestimating the effects of the disease, as has been done in other similar studies^{24,25,27}."

References cited in this comment:

- Bourne, D. et al. 2022. Diseases of scleractinian corals. in *Invertebrate Pathology* (eds. Rowley, A., Coates, C. & Whitten, M. M. A.) 77 (Oxford University Press).
doi:10.1093/oso/9780198853756.003.0004
- Dahlgren C et al. 2021. Spatial and Temporal Patterns of Stony Coral Tissue Loss Disease Outbreaks in The Bahamas. *Front. Mar. Sci.* 8:682114. doi: 10.3389/fmars.2021.682114
- Gintert, B. E. et al. 2019. Regional coral disease outbreak overwhelms impacts from local dredge project. *Environmental Monitoring and Assessment* 191, 1–39.
- Miller, M. W. et al. 2016. Detecting sedimentation impacts to coral reefs resulting from dredging the Port of Miami, Florida USA. *PeerJ* 4, e2711.
- Muller, E. M. et al. 2020. Spatial Epidemiology of the Stony-Coral-Tissue-Loss Disease in Florida. *Frontiers in Marine Science* 7, 11.
- Neely, K. 2018. Surveying the Florida Keys Southern Coral Disease Boundary. Florida DEP. Miami, FL. Pp. 1-15.
- Precht, W. F. et al. 2016. Unprecedented Disease-Related Coral Mortality in Southeastern Florida. *Scientific Reports* 6, 1–11.
- Raymundo, L. et al. 2008. *Coral Disease Handbook Guidelines for Assessment, Monitoring and Management*. Management.

Lines 401-403: See above for a solution to your *Siderastrea* problem.

R: Thank you. This is not really a problem, we just wanted to clarify that this species was considered as being affected by SCTLD. This is relevant information when comparing our results with those of earlier studies conducted in Florida that considered it to be a different disease (e.g., Precht et al., 2016). However, microbiome studies have now shown that diseased *S. siderea* colonies show a consistent microbial signature when compare with those of other afflicted species (Clark et al., 2021).

References cited in this comment:

- Clark, A.S. et al., 2021. Characterization of the Microbiome of Corals with Stony Coral Tissue Loss Disease along Florida's Coral Reef. *Microorganisms* 2021, 9, 2181.
- Precht, W. F. et al. 2016. Unprecedented Disease-Related Coral Mortality in Southeastern Florida. *Scientific Reports* 6, 1–11.

Results

Line 108-Consider moving Fig. S1 as a main figure and moving the PCA plots as supplementals. Figure S1 nicely illustrates demographic effects of SCTLD.

R: Thank you for this comment. We are pleased that you liked Figure S1. However, as mentioned in an earlier comment, Figure S1 and Figure 3a-c do not show the same information, and we cannot replace one with the other.

Figure S1 only shows the densities of colonies for each species before and after the outbreak, lumping all sites (i.e., there is no information at the site level or at the coral community level). In contrast, the Principal Component Analyses (PCA) in Figure 3a-c show the changes in composition at the site level; the absolute contributions of species, families, or traits to the observed differences; and the 95% confidence intervals that show how different the groups are (i.e., 'pre-outbreak' vs 'post-outbreak'). These ordination plots allowed us to represent the drastic transformation towards more homogenous assemblages, as determined by taxonomic and functional trait data, with a notorious lack of contributions from the most severely afflicted species during the post-outbreak period (Fig. 3a-c).

Line 131-132: I would think too an important distinction with Banco Chinchorro is absence of human development there, correct (at least compared to Cozumel)? Might want to add that to narrative if true.

R: This is a very good point. We have now clarified that there is minimal human use of Banco Chinchorro. This location only hosts fisher campsites and research and military stations.

Lines 241-243: "However, the resulting wide-spread coral mortality described here was dictated by the vulnerability of species to SCTLD, and thus caused non-random changes in community structure that further and radically affected the functional integrity of the coral communities." I don't understand how that makes SCTLD so special. After all, the widespread decline of Acroporids in the late 1970s-80s were dictated by vulnerability of that particular family to unexplained tissue loss. How is that any different than SCTLD?

R: Thank you for this comment. We did not mean to imply that SCTLD is more special than the disease that affected the acroporids. On the contrary, we aimed to emphasize the

historical context and status of coral reefs before the SCTL D outbreak in this paragraph. However, it is important to highlight that the main difference between SCTL D and previous disease outbreaks is the large number of susceptible species, many of which share specific traits that disproportionately affect the contributions of a single morpho-functional group.

We have reworded some parts of this paragraph to improve clarity:

“The ecology and physical functionality of coral assemblages in the Caribbean were undergoing severe ecological changes prior to the SCTL D outbreak. Chronic and acute disturbances had progressively driven a decline in the abundance of the main reef-building corals that was accompanied by a concomitant increase in the relative or absolute abundance of opportunistic species characterized by small-sized colonies that do not notably contribute to reef structure and are known to be tolerant to environmental stress (Fig. 5; ^{9,32}). The pre-SCTL D communities were described as ‘shifted’ coral assemblages, and the contributions of formerly dominant acroporids were often negligible given their reduced abundance, whereas large massive species remained and contributed the most to ecosystem structure and functionality (Fig. 5; ^{32,41-43}). However, the resulting wide-spread coral mortality described here was dictated by the vulnerability to SCTL D of some species that share key morpho-functional traits (Fig. 1a), and thus caused non-random changes in community structure that further and radically affected the functional integrity of the coral communities.

The morpho-functional groups comprised of large and massive species were the most afflicted by the SCTL D outbreak (Fig. 1), whereas the species mildly affected by the disease showed relative increases in abundance. The post-SCTL D coral communities are now represented by a hyper-domination of opportunistic corals, although this remarkably seems to be accompanied by an apparent resurgence of acroporids as key functional elements (Figs. 4a-c, 5)...”

Line 249: I am curious as to definition of an opportunistic coral. For instance, could fast growing Acropora be called opportunists too?

R: Thank you for pointing this out. We have expanded the definition of ‘opportunistic’ as follows:

"species characterized by small-sized colonies that do not notably contribute to reef structure and are known to be tolerant to environmental stress". Please see the previous comment for context.

Lines 262-274: Lots of jargon to unpack here. How do functions serve to "track" increase sea level? Why is low calcification in other regions of the world relevant to your study site? Can you provide a citation to show that calcification rates were already low prior to SCTL D in Caribbean? That would be more germane. How does reef construction "track" sea level rise? What does tridimensionality mean? Seems all of this could be condensed to a single sentence as follows: "Loss of corals to SCTL D will lead to further bioerosion of coral reef ecosystems making coastal communities more susceptible to sea level rise caused by climate change."

R: Please see our response to your next comment.

Lines 300-318: You touch on similar themes as above. Suggest you condense these 2 paragraphs into a couple of succinct sentences explaining why loss of corals will lead to bioerosion and increased coastal hazards due to climate change." I think that is what you are trying to communicate.

R: Thank you for the comment. We agree that the text mentioned in this comment and in the previous comment was a little redundant. As suggested, we have simplified the language and reduced the amount of text. We believe that these changes have aided us in concisely and directly delivering the main ideas we wanted to communicate.

The amended version of the text is now presented in one paragraph in Lines 277-295:

"The large-scale loss of the functionally important corals defined radical shifts in reef conditions and dynamics, exacerbating further losses of ecological integrity along the entire reef track. The outcomes of coral die-off from the SCTL D outbreak will compromise key functions that are supported by living reef-building corals such as reef framework production, the maintenance of reef habitat complexity, and the potential for growth⁹. These functions largely depend on the capacity of coral assemblages to accumulate calcium carbonate at higher rates than the rate of loss due to biological, chemical, or physical erosion. If erosive processes equal or exceed reef carbonate production, reef frameworks may be destroyed faster than they are produced, resulting in a net negative carbonate

budget^{48,49}. In this study, we observed a nearly 30% reduction in the capacity of coral communities to produce calcium carbonate. This is alarming because levels of community calcification prior to the impacts of SCTLD were substantially below the optimal rates that have been reported under high coral cover states in the Caribbean^{49,50}. In the absence of recovery, the ultimate consequences of coral mortality will thus be modulated by destructive forces like bioerosion or the biogenic dissolution of reef structures⁵¹. This is particularly relevant as both bioerosion rates and skeletal dissolution are thought to become pervasive when the water chemistry changes or the temperatures increase^{48,51}. Our understanding of how the increased availability of substrate for bioeroders will interact with rapid environmental changes remains limited. However, if the ultimate objective is preserving coral reef functioning and services, it may be necessary to focus on replenishing and favoring the recovery of coral communities while improving our understanding of how to control and modulate the destructive forces operating within coral reefs."

Line 333: I think you mean here allee effect (2).

R: Yes, this was a typo. Changed to "*Allee effect*".

Figures

Figure 1. Consider deleting C-E. There are now plenty of photos of SCTLD in literature, and these do not add substantially to your message.

R: Thank you for the suggestion. We, however, prefer to keep these photos in the main document. This manuscript is aimed at a broad audience and not only at reef scientists working with coral diseases. We believe these images are helpful as visual references for diseased and recently dead corals (both used in our study; please see the Methods).

Figure 2. Delete Panel A. See my comments re. white plague. Also delete violin plots. By promoting Figure S1 as a main figure, you illustrate there very well effects of SCTLD on coral demographics. This then simplifies the figure to single panel (B).

R: Thank you for the suggestion. However, as explained in our response to your comment about lines 375-377 regarding White Plague, Figure 2a (which shows no evidence of white plague-type diseases in the pre-SCTLD period) is an important element, as it allows to support (1) that the outbreak started in 2018 and (2) that the outbreak is the main cause of the observed changes. Please also note that when comparing Figures 2a and 2b, it is

possible to observe the drastic decline in susceptible species in the 'post-SCTLD' period (symbols are consistently smaller in Figure 2b compared to those in Figure 2a).

Also, as mentioned in your comment regarding Line 108, Figure S1 and Figure 3a-c do not show the same information, and we cannot replace one with the other. Therefore, we decided not to move Figure S1 to the main document.

Figure 4. Delete. I'm unconvinced that this figure means anything. It is basically a cartoon trying to illustrate that corals have declined over time in the Caribbean and that relative proportions of different corals have changed over time. All of that has been well documented in literature with concrete survey data (Cramer et al., Sci. Adv. 2020; 6 : eaax9395). You can cite those studies to make your point without having to resort to a cartoon. Plus, deleting it will simplify your central message.

R: Thank you for the suggestion. Yes, this conceptual figure shows how the relative proportion of different corals has changed over time. Although the figure is a simple representation of the history of Caribbean coral communities, we believe it clearly summarizes most of the literature in this sense (including the SCTLD outbreak). In addition to our findings, this figure compiles the results of 16 other studies, which are included in Table S4.

We consider that the figure is relevant and have decided to retain it for two main reasons:

(1) It shows that the apparent increase in acroporids is primarily an artefact of the drastic reductions in the relative contributions of many other species due to SCTLD. Without this figure, it would be easy to mistakenly believe that we are reporting that acroporids have increased or undergone a resurgence. Please see our response to your comment about Lines 32-34.

(2) It provides a visual representation of the historical context of Caribbean coral communities before SCTLD. Please see our response to your comment about Lines 241-243. As you mentioned, other threats have affected and drastically changed Caribbean reefs. We want to be very clear about this and frame our results in a broader context considering several decades of change.

Supplemental

Table S1 title: "...significant p values." Also, I do not see bolded p values in that table.

R: Amened.

References

- 1) Alvarez-Filip, L, NK Dulvy, JA Gill, IM Cote, and AR Watkinson. (2009). Flattening of Caribbean coral reefs: region-wide declines in architectural complexity. *Proceedings of Royal Society B* 276:3019–3025.
- 2) *Journal of Experimental Zoology*. 61 (2): 185–207.

REVIEWERS' COMMENTS:

Reviewer #1 (Remarks to the Author):

I would like to thank the authors for their (continued) willingness to make changes to their statistical modeling approaches. They have addressed all of my statistical concerns, particularly those related to modeling disease prevalence. I have no further comments.